# Dynamics of Endothelial Cell Diversity and Plasticity in Health and Disease

**DOI:** 10.3390/cells13151276

**Published:** 2024-07-29

**Authors:** Alexey Larionov, Christian Manfred Hammer, Klaus Fiedler, Luis Filgueira

**Affiliations:** 1Faculty of Science and Medicine, Anatomy, University of Fribourg, Route Albert-Gockel 1, CH-1700 Fribourg, Switzerland; christian.hammer@unifr.ch (C.M.H.); luis.filgueira@unifr.ch (L.F.); 2Independent Researcher, CH-1700 Fribourg, Switzerland; contact@klausfiedler.ch

**Keywords:** endothelial cells, microcirculation, macrocirculation, development, WNT, VGF, NOTCH, BMP, TGF, EMT, EndMT, cancer, endothelial turnover, endothelial regeneration

## Abstract

Endothelial cells (ECs) are vital structural units of the cardiovascular system possessing two principal distinctive properties: heterogeneity and plasticity. Endothelial heterogeneity is defined by differences in tissue-specific endothelial phenotypes and their high predisposition to modification along the length of the vascular bed. This aspect of heterogeneity is closely associated with plasticity, the ability of ECs to adapt to environmental cues through the mobilization of genetic, molecular, and structural alterations. The specific endothelial cytoarchitectonics facilitate a quick structural cell reorganization and, furthermore, easy adaptation to the extrinsic and intrinsic environmental stimuli, known as the epigenetic landscape. ECs, as universally distributed and ubiquitous cells of the human body, play a role that extends far beyond their structural function in the cardiovascular system. They play a crucial role in terms of barrier function, cell-to-cell communication, and a myriad of physiological and pathologic processes. These include development, ontogenesis, disease initiation, and progression, as well as growth, regeneration, and repair. Despite substantial progress in the understanding of endothelial cell biology, the role of ECs in healthy conditions and pathologies remains a fascinating area of exploration. This review aims to summarize knowledge and concepts in endothelial biology. It focuses on the development and functional characteristics of endothelial cells in health and pathological conditions, with a particular emphasis on endothelial phenotypic and functional heterogeneity.

## 1. Introduction

ECs are the most ubiquitous cells of the body, with a wide variety of functions. The total surface area of the ECs in the human body is approximately 3000–6000 m^2^, corresponding to 1 to 6 × 10^13^ cells [1]. The endothelium is a multifunctional tissue, an internal organ, that performs protective (hematoencephalic, intestinal, renal, and hepatic barriers), receptor, transport (oxygen, nutrients, growth factors), secretion, absorption, and synthesis functions [2]. It is also involved in general and local tissue homeostasis, vascular tone and blood pressure regulation, hemostasis, angiogenesis, innate and adaptive immune responses, inflammation, vascular remodeling, tumorigenesis, and initiation of cardiovascular diseases [3,4,5,6,7].

ECs are classified within the simple squamous epithelium group, reflecting their arrangement in a single layer of flat cells. Although the EC population demonstrates heterogeneity across different organs, it shares unique histological characteristics within the same vascular bed region and originates from a common embryonic source. Regularly manifested unifying morphological endothelial features include:(1)A flat and elongated teardrop or cobblestone-like shape with a polygonal cell outline. The endothelial shape facilitates the blood flow dynamics (laminar blood flow due to the ECs not being studded with ciliae), and it is crucial for selective barrier formation and adaptation to environmental changes [8,9,10].(2)Possession of different junction types. The junctional complexes play a crucial role in the formation and stabilization of the endothelial barrier and in regulating permeability between the blood and surrounding tissues [11,12,13,14,15]. Barrier specialization is directly related to junction types that specifically affect the endothelial layer’s permeability, endothelial cell growth, apoptosis, and intracellular signal transmission [16,17]. Cell–cell junctional contacts impact the rearrangement of the endothelial layer through different signaling pathways (e.g., PI3Ka/MYPT1/MLCP [18], Notch, Rho GTPase, Wnt/beta-catenin, and Hippo pathways [19,20,21,22];(3)The presence of a strong extracellular matrix basal lamina, reinforcing the ECs and contributing to their trophic functions, such as survival, proliferation, and differentiation [23,24,25]. The basal lamina is associated with the vessel networks that are essential for maintaining local tissue homeostasis, post-transcriptional modifications, and possibly even the regulation of gene expression [26];(4)Polarity of the endothelial cell surface. Herein, we refer to endothelial asymmetry in the structural organization and components of the apical and basolateral surfaces of ECs. This property is one of the keys to maintaining the endothelial barrier, facilitating cell migration and vectorial transport of biomolecules and enabling proper signaling. The polarity is closely related to the positioning and distribution of the centrosomes and Golgi apparatus, centrosomal microtubule proteins (Atp6ap2, Tacc3-ch-Tog, Cep41, CG-Nap), and non-centrosomal microtubule proteins (Campsap2, Par-6, Pkc3) [27,28,29,30,31,32,33,34,35,36], as well as to the sorting of proteins and asymmetric protein surface distribution [37].

Plasticity is another vital characteristic related to the reversible modification of the endothelial phenotype. This quality is multifaceted and based on the complex molecular interplay between epigenetics and genetics. Adaptational changes entail different mechanisms, including the endothelial-to-mesenchymal transition process (EndMT) [38,39], changes in transcription factors released from chromatin versus altered import or affected “extra-local“ synthesis processing [40,41], feedback to environmental cues [42,43], and metabolic modulation [44,45].

Plasticity and heterogeneity are fundamental endothelial features that determine the role of ECs in various physiological processes and predispose endothelia to participate in numerous pathological tissue alterations. Plasticity, as a predictor of endothelial heterogeneity, enables ECs to adapt their phenotype and function to the specific local requirements of their microenvironment. A complex interplay between the local microenvironmental signals and epigenetic cues regulates the unifying EC characteristics mentioned earlier. ECs are inherently changeable, which underpins their heterogeneity. The specialization and function of ECs, regional variation in blood supply, and unique yet differing endothelial cell neighborhoods in different organs dictate the distribution of distinct endothelial cells and their associated phenotypes throughout the organism [46].

This review seeks to illuminate the role of endothelial plasticity as a predictor of heterogeneity in vascular health and disease. Its actuality lies in an attempt to expose the key mechanisms behind endothelial heterogeneity and adaptation under physiological and pathological conditions.

## 2. Endothelial Development

### 2.1. Endothelial Origin, Cell Specification, and Vessel Organization

ECs are mesodermal derivates [47,48]. Endothelial development is closely connected to hematopoiesis and includes three consecutive waves of cellular formation. The first transient pool (or first wave) of ECs appears during extraembryonic hematopoiesis in the yolk sac. Between days 12 and 15 of gestation, the human yolk sac’s hemangioblasts start forming blood islands [49]. The blood islands consist of internal and external layers [50]. The external layer gives rise to the angioblasts, the multipotent progenitors of ECs, while the internal layer contains hematopoietic precursors of the future blood cells [51]. The second wave (extraembryonic hematopoiesis) includes the migration of extraembryonic precursor cells into the embryo and the formation of intraembryonic hemogenic endothelial cells (IHECs) in the aorta–gonad–mesonephros and liver [52]. The third wave (intraembryonic wave) is characterized by the production of hematopoietic stem cells from the IHECs and further colonization of the hematopoietic organs, e.g., aorta–gonad–mesonephros region, liver, and bone marrow [53]. The embryonic ECs do not demonstrate any conventional, fundamental functions (protective, signal transmission, and interstitial regulation), but take part in the structural organization and reorganization of the vascular system, including vasculogenesis (de novo formation of primitive blood vessels from angioblasts), angiogenesis (growth and development of new vessels from the pre-existing vascular network), and remodeling of the existing vessels [54]. Vasculogenesis is characterized by the mobilization and specification of the angioblasts with further differentiation into endothelial cells and proliferation, migration, coalescence, and formation of vascular plexuses (Table 1) [55,56,57,58,59,60,61,62,63,64,65,66,67,68,69,70,71,72,73,74,75,76].

Angiogenesis is a more general physiological process occurring not only in embryogenesis, but also in postnatal life. Arterial and venous angiogenesis can include various processes (endothelial sprouting and intussusceptive microvascular growth) that include morphological changes of the initial vascular plexuses, resulting in vascular network establishment (Table 3) [77,78,79]. In postnatal life, vascular network formation mainly occurs through angiogenesis (including arterio/venogenesis) or remodeling of pre-existing vessels, including collateral vessel formation [80].

Lymphatic vessel development (lymphangiogenesis) from the vein endothelium typically happens in prenatal life. In many studied cases, postnatal lymphatic vessel formation is pathological (Table 2) [58,65,72,81,82,83,84,85,86]. In 2004, a new scheme of the differentiation of the lymphatic vasculature in prenatal life was proposed by G. Oliver [87]. According to him, prenatal development of lymphatic vessels includes four phases: (1) lymphatic ECs competence (ability to respond to stimuli and initiation of LYVE 1 expression); (2) lymphatic ECs bias (formation of lymphatic structures with the involvement of Prox1 transcription factor); (3) lymphatic ECs specification (initiation of specific lymphatic marker expression, e.g., podoplanin (T1α) and neuropilin2 (NRP2); and (4) lymphatic vessel maturation (spreading of the lymphatic vessels in the body) (Table 2) [87,88]. Lymphangiogenesis is crucial for the embryo’s healthy development, clearance of interstitial metabolites, and circulatory liver metabolite supply in the adult, as well as for wound repair and regeneration of the uterine mucosa during the menstruation cycle [67,78]. Tumor metastasis, moreover, is often critically dependent on the lymphatic conduits as well as on blood vascular remodeling [89,90].

**Table 2 cells-13-01276-t002:** Morpho-functional classification of lymphatic ECs.

Lymphangiogenesis Growth and Development of Lymphatic Vessels in Prenatal and Postnatal Life		
Mechanism and phases	Signaling and transcriptional regulators	Markers of endothelial differentiation
I. Prenatal lymphangiogenesis	1. ETS domain protein [84] 2. SOXF factors: SOX7, SOX17, and SOX18 [81,83,85,86] 3. Vascular endothelial growth factor-C (VEGF-C)/vascular endothelial growth factor-F (VEGF-F)/vascular endothelial growth factor-D (VEGF-D) [59,81,82,83,84,85,86,87,88] 4. Prox1 [58,81,83,84,85,86] 5. Forkhead box C2 (FOXC2) [81,82,84,85,86] 6. Rho family GTPase (RAC-1) [85] 7. Tyrosine kinase Syk [81,85] 8. SLP 76 [81] 9. Phosphatase-Cγ2(PLCγ2) [84] 10. Semaphorin 3F (SEMA3F) [81,84] 11. Chicken ovalbumin upstream promoter transcription factor (COUP-TFII) [58,83,85]	Platelet endothelial cell adhesion (PECAM-1) [84,85] CD34 [80,81,84,87] CXCR4 (through CXCL12 stimulation) [58,85] LYVE-1 [80,82,83,84,85,86,87] Podoplanin [58,80,81,84,87] Vascular endothelial growth factor receptor 3 (VEGFR-3) [80,81,82,83,84,85,86,87] PROX-1 (Prospero homeobox protein) [58,81,83,84,85,86] CD44 [80,87]
(1) Classical Lymphatic vascular development: centrifugal sprouting from primary lymph sacs arises from embryonic cardinal veins (starts at E9.5-10.5 in mice and 6–7 weeks in humans)		
A. Budding and sprouting lymphatic ECs from cardinal veins [82,83,84,85,86]		
B. Centrifugal migration lymphatic ECs [82,83,84,85,86]		
C. Proliferation lymphatic ECs and generation of a one-way network of capillaries [82,83,84,85,86]		
(2) Lymphatic vascular differentiation, according to G. Oliver (2004) [87]		
A. Lymphatic competence (lymphatic ECs from a vein at the E9.0-9.5 receive the ability to answer specific lymphatic-inducing signals) [87]		
B. Lymphatic bias (determination of the lymphatic ECs fate, approx. E9.0-10.5) [87]		
C. Lymphatic specification (lymphatic ECs differentiate into the desired phenotype independently from microenvironmental cues, approx. E10.5.-14.5) [87]		
D. LEC differentiation (maturation and separation of lymphatic vessels, approx. E14.4. postnatal life) [87]		
II. Postnatal lymphangiogenesis		
*closely related to such pathological processes as implantation and tumorogenesis (phases similar to angiogenesis)*		
A. Sprouting lymphangiogenesis: occurs with or without lymph flow in pre-existing vessels [58]		
B. Intussusceptive lymphangiogenesis is dependent on lymph flow [58]		

### 2.2. Pathways and Factors Involved in the Regulation of Endothelial Development

Diversified signaling pathways and transcription factors regulate both vasculogenesis and angiogenesis (Table 1), including WNT (wingless), NOTCH (“notched wings”), bone morphogenetic protein (BMP), fibroblast growth factor (FGF), and vascular endothelial growth factor (VEGF) [91]. A wide range of transcription factors participates in the regulation of endothelial development, including the E twenty-six family of transcription factors or ETS family (ETS1, ETS2, FLI 1, ERG, ETV1, ETV2, ELK3); GATA transcription factors; VEGF-A, vascular endothelial growth factor receptor 2 (VEGFR2, Flk1/KDR); stem cell leukemia/or T-cell acute lymphocytic leukemia-1 (SCL/TAL1); SMADS proteins; BMP; Lim domain only 2 (LMO2); retinoic acid; and transforming growth factor β (TGFβ) [46,92,93,94].

WNT signaling belongs to the ancient pathway responsible for endothelial development in embryogenesis and consists of canonical (β-catenin associated signaling) and non-canonical (β-catenin independent, including planar cell polarity (PCP) and WNT/Ca2+ signaling) pathways [95]. The canonical and non-canonical pathway initiation starts with interactions of the extracellular WNT ligand with the N-terminal domain of one of the seven transmembrane-span frizzled receptors (FZD receptors) [96,97]. The WNT signaling pathway plays an essential role in embryonic vascular development and the pathogenesis of a wide range of diseases, including cancer, osteoarthritis, muscle pathologies, and kidney diseases [98,99,100,101,102]. The WNT/β-catenin-associated pathway is a primary regulator of brain development, neural differentiation of pluripotent stem cells, and neurodegenerative disorders [103]. WNT/β-catenin signaling, as reported by transcription factor 7-like 2 (TCF7L2)-dependent transcription, is involved in the onset of mental disorders and behavioral deficits [104]. WNT signaling participates in angiogenesis through interaction with secreted frizzled-related protein (SFRP), which is vital for vascular growth. SFRP1 is expressed by all cultured endothelial cells and favors vascular development in all angiogenic models, whereas SFRP2 is mainly involved in angiogenic responses through the nuclear translocation of nuclear factors of activated T cells (NFATs) [105]. SFRP1 and SFRP2 are agonists of the WNT signaling pathway, while SFRP4 appears to antagonize the canonical WNT signaling [106].

NOTCH signaling is an intracellular pathway that controls cell-type specification, differentiation, proliferation, regeneration, and tissue development [107]. NOTCH regulates cell fate, switching in a growing sprout from tip to stalk (meaning initiation “endothelial nascent sprout” from tip and stalk endothelial cells) [108] in the cardiovascular system and the vascular tree. It maintains the proper arterial, venous, and capillary organization; endothelial phenotypic identity is proposed to be regulated by shear stress influence on the NOTCH1 mechanosignaling cascade in ECs [109]. Moreover, the postnatal lymphangiogenesis and differentiation from veins occur under NOTCH signaling control [110]. Its principal components are ligands, receptors, DNA binding protein, and downstream transcribed genes (regulated expression) [111]. NOTCH is composed of four transmembrane receptors (NOTCH 1,2,3,4) and NOTCH ligands (JAGGED1, JAGGED2, DELTA-LIKE 1,2,3, SERRATE). The receptor–ligand interaction drives the NOTCH receptor’s proteolytic cleavage with subsequent liberation of the NOTCH intracellular domain (NICD) into the cytoplasm, activating canonical and non-canonical signaling mechanisms [112]. The canonical pathway is characterized by the migration of the NICD into the cellular nucleus, where it regulates hairy and enhancer of split-1 (HES1) and hairy/enhancer-of-split-1 related to YRPW motif protein 1 (HEY1) target genes by forming transcriptional complexes with the recombination signal-binding proteins for immunoglobulin kappa J region (RBPJ) and mastermind-like protein 1,2,3 (MAML1,2,3). In the non-canonical pathway (RBPJ-independent NOTCH signaling), NOTCH is activated through interaction with IKKA (IkappaB kinase α) of the NF-κβ (nuclear factor kappa-light-chain-enhancer of activated B cells) pathway, with lymphoid enhancer-binding factor 1 (LEF1) of the WNT signaling, or with R-Ras, which is responsible for cell adhesion [113,114,115].

BMP is part of the TGF-β superfamily, participating in the regulation of cardiovascular and lymphatic development [116]. TGF-β regulates the interaction between ECs and the microenvironment (vascular smooth muscle cells, pericytes). TGF-β takes part in embryonic vascular development, vascular integrity, endothelial activation, and proliferation [117]. BMP activates SMAD-dependent (canonical) and multiple SMAD-independent (non-canonical) pathways. Interaction of BMP with cell surface receptors leads to the formation of heterotetrameric complexes. Type I and type II serine/threonine kinase receptors change the gene transcription and signal transduction cascade [118]. The type I receptors include activin receptor-like kinase-1,2,3,5,6. The type II receptors are transmembrane serine/threonine kinase receptors, which are constitutively active and include BMP2, type 2 activin receptor (ACTR-2A), and type 2B activin receptor (ACTR-2B) [119]. The canonical BMP signaling pathway implicates phosphorylation of SMAD1 and SMAD5 intracellular mediators responsible for initiating angiogenesis in venous and arterial ECs. Notably, the conditional knockout of SMAD1/5 in the ECs results in the downregulation of NOTCH signaling. Disruption of SMAD1/5 is the cause of abnormal vascular development and defective lineage specification of ECs [120,121]. Non-canonical BMP signaling occurs by activating TAK-1 (transforming growth factor-β activated kinase 1), PI3K/Akt (phosphatidylinositol-3 kinase/protein kinase B), and Rho-GTPases [122].

The FGF signaling interacts with transmembrane tyrosine kinase receptors, activating their pathways (PI3K/Akt, RAS/MAPK (RAS superfamily/mitogen-activated protein kinase), and phospholipase Cγ/PLCγ). The FGF protein family contains 22 signaling ligands. However, only FGF1 (acidic FGF) and FGF2 (basic FGF) are crucial for angiogenesis [123]. FGF signaling is necessary for developing ECs, injury response, repair, and the initiation of pathologies (cancer, abnormal angiogenesis) [124]. Interestingly, FGF-2 can modulate the endothelial expression of VEGF through both autocrine and paracrine mechanisms of action [125].

The ETS transcription factors are expressed in embryonic endothelial cells and stimulate initial capillary network formation through proliferation, cell spreading, and adhesion. A lack of ETS factors provokes severe defects in embryonic angiogenesis [94,126,127].

GATA is a zinc-finger family of transcription factors essential for developing the endocardium and vessels [128]. GATA-4,5,6 are mostly expressed in the cardiac endothelium, while GATA2 plays a critical role in blood cell development [129].

VEGF is a potent angiogenetic factor for the proliferation and migration of ECs during angiogenesis and vasculogenesis [130]. VEGF binds to VEGFR: VEGFR1 (or FLT1) and VEFR2 (or FLK1). FLK1 mediates almost all known angiogenesis-dependent functions (migration, survival, vascular permeability) [131,132,133]; the additional function of Flt1 relates to organ differences in physiological and pathological conditions such as angiogenesis [134], homeostasis, preeclampsia, and metastasis [135,136]. In this sense, VEGF-A/Flk1 are the earliest markers of mesodermal differentiation and potent angiogenesis promoters, responsible for physiological and pathophysiological permeability, vascular organization, cell cycling, and differentiation [137,138,139].

The SCL/TAL1 factor is a transcription factor with a helix–loop–helix structure, regulating hematopoiesis and endothelial cell reprogramming [140]. SCL/TAL1 stimulates transcription by involving a core complex on DNA containing E2a/HEB, GATA1-3, LIM domain only 2-rhombotin-like 1 (LMO1/2), LIM-only, LIM domain-binding 1 (LDB1), ETO2, runt-related transcription factor 1 (RUNX1), and FLI1 [141]. SCL/Tal-1 is essential for initiating mesodermal endothelial differentiation, vascular network formation, and postnatal angiogenesis [142].

The LMO2 transcription factor is involved in vascular remodeling through the LIM-domain zinc-finger-like structures and DNA interaction [143]. The deficiency of the LMO2 results in a decrease in endothelial migration [144].

Retinoid acid contributes to endothelial differentiation. The lack of retinoid acid leads to embryonic death, with multiple defects in vessel reorganization of the yolk sac [145]. The retinoid acid receptor is essential for angiogenesis regulation [146], EC proliferation [147], and blood barrier regulation [148].

The control of endothelial development is complex and context-dependent (Figure 1A) [149,150,151,152,153,154,155,156,157,158,159,160,161].

The antagonistic, agonistic (reciprocity), and synergistic interplays between the pathways result from combinations of intrinsic and extrinsic cues: type and subtype of pathways (e.g., NOTCH differs from 1 to 4), tissue specificity, timing of pathway activation (simultaneous or sequential), duration of signaling, impact of other pathways, etc. (Figure 1B) [153]. BMP and NOTCH demonstrate antagonistic effects [162,163]. BMP/WNT/TGF-β demonstrate synergetic regulation and determine the ligand production of each other; NOTCH acts downstream of VEGFR; and WNT-NOTCH cross-talk has been shown to regulate multiple effects on cell fate. WNT plays the role of agonist and antagonist for NOTCH in cardiogenesis, and WNT and TGFβ, in most cases, demonstrate an agonistic relationship in extracellular signaling [153,164,165,166].

## 3. Endothelial Ontogenesis

### 3.1. ECs and Stem Cells Interchange

During postnatal ontogenesis, a dynamic reciprocal exchange between the stem cells and ECs is vital for vascular regeneration and the maintenance of physiological and morphological integrity. Continuous endothelial regeneration relies on the ability of stem cells to be converted into ECs. There are different types of stem cells that demonstrate strong endothelial potential: human embryonic stem cells [167], adult stem cells [168], and hematopoietic stem cells [169]. The postpartum physiological (wound healing) and pathological (tumor angiogenesis) endothelial renewal occurs through the migration of pre-existing ECs from the vascular wall or via recruitment of endothelial precursors, so-called endothelial progenitor cells, from the circulatory system [170,171]. Endothelial progenitor cells are the provasculogenic subpopulation of hematopoietic stem cells, demonstrating a profound capacity for proliferation, specialization, differentiation, and maturation [172,173,174]. Endothelial progenitor cells are subdivided into erythro-myeloid progenitors, non-hematopoietic endothelial progenitor cells [175], endothelial colony-forming cells, or hematopoietic endothelial progenitor cells [176]. The primary source of non-hematopoietic endothelial progenitor cells is blood [177,178]. Bone marrow-derived hematopoietic endothelial progenitor cells can be considered as a provasculogenic population of hematopoietic stem cells. The hematopoietic stem cells and endothelial progenitor cells express specific surface markers such as CD34, CD133, CD105, FLK-1/KDR, and CXCR4 [179]. In humans and animals, the bone marrow niche is formed by specialized tissue with a specific microenvironment where cellular and molecular components regulate the cell fate and behavior of endothelial progenitor cells [180]. Bone marrow includes vascular (hematopoietic stem cells, ECs, mesenchymal stem cells, and endosteal cells (osteoprogenitor cells and osteoclasts under the control of parathormone)) niches [181]. The functional regulation of the bone marrow niches, determining the cell fate, mostly occurs via epigenetic cues: epigenetic mechanisms (histone modifications, DNA methylation, non-coding RNAs) [182], sympathetic and parasympathetic innervation [183], and metabolic and chromatin modifications [184].

### 3.2. Role of ECs in Hematopoiesis and Vasculogenesis

Hematopoiesis is the formation of blood cells, which includes primitive (in the yolk sac) and definitive (aorta–gonad–mesonephros region, liver, bone marrow) stages [185]. Vasculogenesis is the de novo formation of blood vessels [186]. During the early stages of embryogenesis, vasculogenesis and hematopoiesis occur in parallel and are closely linked to each other [187]. In many mammalian species, hematopoietic stem cells, hematopoietic precursors, develop directly from hemogenic endothelium [188]. Hemogenic ECs are genetically programmed to transdifferentiate into hematopoietic stem cells through endothelial-to-hematopoietic transition and EndMT [189,190]. The cooperation of EC and hematopoietic progenitor stem cells is indispensable for the formation of blood islands (hematopoietic precursors, hematopoietic progenitors) in the yolk sac, fusion, and development of the primitive capillary plexus [191]. Endothelial specification occurs under the control of the well-defined and aforementioned signaling pathways: BMP4, WNT [61], NOTCH, and Hedgehog (Hh) [192]. The Hh signaling pathway is involved in the control of embryonic and adult hematopoiesis and is required for endothelial patterning; Hh mutants demonstrate a functional lack of vascular remodeling [193]. A number of cytokines and interleukins (granulocyte-colony stimulating factor (GM-CSF), IL-3, IL-4, IL-6, IL-8) regulate hematopoiesis and contribute to endothelial growth and differentiation as well [194]. VEGF, angiopoietin-1 [195], FGF-2, TGF-α, TGF-β, HGF, tumor necrosis factor-α (TNF-α), angiogenin [194], and recombinant human erythropoietins (EPO-α and EPO-Z) [196] stimulate vasculogenesis. Moreover, the endothelial and hematopoietic fate is regulated by TAL-1 (also known as SCL), GATA-1 and 2, LMO2, RUNX-1 [188], and stromal cell-derived factor-1 alpha (SDF-1 alpha) [197]. The endothelial cell-selective adhesion molecule (ESAM) is a member of the immunoglobulin superfamily. It facilitates communication between ECs and determines interactions between ECs and hematopoietic stem cells in hematopoiesis and vasculogenesis [198,199,200].

## 4. Endothelial Plasticity as the Base of Endothelial Diversity

All endothelial cells are multivalent players and messengers contributing to local and organism-wide functions. ECs exhibit structural and functional plasticity, leading to diversity in the endothelial population.

Heterogeneity is determined by a combination of (1) different genes [201], e.g., homeobox genes (HOX family of transcriptional factors B3, D4) [202], E-selectin, thrombomodulin, von Willebrand Factor (vWF), platelet endothelial cell adhesion (PECAM-1), and Ephrin B4 genes [203]; and (2) epigenetic cues, e.g., physical influences (low blood flow versus high blood flow), intrinsic biochemical characteristics (regulatory proteins, hormones, transcription factors), non-coding RNAs, and functional and morphological biochemical mutual interactions [204].

The genes and receptors that are sensitive to environmental changes include neutrophin-4, transforming growth factor-β, VEGFR-3, interleukin-1 receptor type 1 precursor, E-selectin, integrin α4 precursor, C cathepsin, cytochrome P450 IVB1, Bad Protein, a myb proto-oncogene, Ku 70-kDa subunit, S100 calcium-binding protein A7, MAL, and a regulator of G-protein signaling 16/a28 [205].

Epigenetic adaptivity and reversibility are due to the modulation of the transcription profile and dedifferentiation of the initial endothelial phenotype [206]. The loss of inductive signals leads to the modification of tissue-specific receptors that affect gene expression, where non-coding RNA regulating methylation and histone modifications play an essential role [39,207]. Physical epigenetic extrinsic factors (inductive signals) such as blood flow, shear forces, hypoxia, and foreign RNA fragments are more powerful regulators. For example, blood circulation with low shear stress alone stabilizes hypoxia-inducible factor 1α (HIF-1α), influences cell reprogramming, and ultimately decreases hypoxia levels [208]. Hypoxic conditions increase VEGF production and also stimulate human fibroblast conversion to ECs [209]. Certain viral Kaposi sarcoma herpesvirus miRNAs (miR-K12-11, miR-K-12-6) delivered via extracellular vesicles, together with transcription factors, can be considered as signaling molecules that can modulate or correspond to the epigenetic environment, or act as effectors of altered environment, participate in cell reprogramming, and stimulate angiogenesis [210,211].

### 4.1. Target Regulation of Endothelial Plasticity and Heterogeneity

Not much is known about the factors that may determine endothelial plasticity.

The pathways thoroughly described previously play a crucial role in this process: NOTCH (generally described previously), WNT (mentioned above), BMP (described previously), TGFβ, (VEGF) [212], Lim domain only 2 (LMO2) [213], ETS variant transcription factor 2 (ETV2) [214], twist family BHLH transcription factor 1 (TWIST-1) [215], and hypoxia-inducible factor 1 (HIF1) [216]. This section specifically focuses on the characteristics associated with endothelial plasticity and heterogeneity.

#### 4.1.1. NOTCH

NOTCH is one of the principal regulators of endothelial heterogeneity, determination, and endothelial apoptosis. NOTCH, as mentioned before, inhibits the proliferation of the sprouting ECs through direct interaction between the receptors and DLL4 [217]. Endothelial plasticity and heterogeneity are closely related to NOTCH 1 and NOTCH 4 receptors. They differently influence endothelial apoptosis and contribute to the complex regulation of vascular remodeling. Thus, NOTCH1 and NOTCH4 determine apoptosis and normal morphology and are related to cellular context and external stimuli. Activation of NOTCH1 signaling leads to an augmentation of cell survival under stress conditions via upregulation of anti-apoptotic proteins Bcl-2 and Bcl-xl [218]. Additionally, NOTCH 1 signaling in cell survival may enhance the expression of survival pathways such as PI3K/Akt [219]. Notch4 signaling is an antagonist of NOTCH1 under certain conditions. Overexpression of NOTCH4 leads to enhanced endothelial apoptosis through the induction of pro-apoptotic factors like p53 and Bax [220]. NOTCH4 leads to the dynamic restriction of vascular networks during development, contributing to the pruning of aberrant vessels [221]. NOTCH signaling participates in the transformation of the venous ECs into artery ECS via the NR2F2 (Nuclear Receptor Subfamily 2 Group F, Member 2) receptor [222].

#### 4.1.2. WNT

WNT is one of the crucial regulators of endothelial plasticity and heterogeneity. As mentioned before, WNT signaling acts 1) all over the body through the canonical WNT/catenin β signaling [223,224,225] and non-canonical pathway WNT/Ca^2+^ [226,227] and 2) locally in context-related responses [228,229]. WNT determines endothelial morphology in a case of disturbed blood flow, controls endothelial integrity, regulates the distribution of tight junctions [223], and controls endothelial maturation [230]. All the aforementioned mechanisms may be considered as the basis of endothelial heterogeneity.

#### 4.1.3. BMP

BMP signaling selectively influences ECs in a context-dependent manner without affecting endothelial survival through SMADS effectors and a non-canonical pathway via sp38, MAPK, and ERK [231]. The transcriptional activation of BMP signaling occurs under distinct epigenetic influences. For example, the turbulent shear stress induces phosphorylation of SMAD1/5/8, which provokes the activation and proliferation of ECs. Laminar shear stress leads to endothelial hibernation through the BMP9/Alki1/ENG-SMAD cascade [232,233]. Interestingly, SMADs may also interrelate with p300/CREB binding protein (CBP) signal transduction and readily create a complex with the NOTCH intracellular domain, activating NOTCH target genes. Post-transcriptionally, BMP targets are related to the reversible phosphorylation process [233,234], ubiquitination, and sumoylation [233,235], affecting cell survival [236,237].

#### 4.1.4. TGFβ

The transforming growth factor-β (TGFβ) signaling pathway plays a vital role in the regulation of EC heterogeneity of the vascular systems. Different aspects of endothelial biology are influenced by TGFβ. This pathway signals through type I and II serine/threonine kinase receptors ALK1 (leading to SAMAD1/5/activation and promotion of the EC proliferation and migration) and ALK5 (phosphorylation of SMAD2/3 leading to inhibition of EC proliferation and migration) [238,239]. The role of TGFβ is that of a potential inducer of EndMT through the SMAD-dependent and SMAD-independent pathways [238,240].

#### 4.1.5. VEGF

VEGF can be considered a driver of endothelial plasticity due to the heterogeneity of ECs. Deletion or mutation of the VEGFR1 has different effects in different organs: in the retina, it increases vascular density, and in the liver, there is no change [231]. Deletion of the VEGFR2 causes regression of the vascularization [231], and VEGFR3, similarly to VEGFR1, initiates hypersprouting of vessels in the retina [241]. It is interesting that different VEGF isoforms determine the phenotypic specialization of endothelial cells in different organs [242]. This explains the phenomenon of ECs of various organs responding differently to VEGF changes in the vascular bed [243].

#### 4.1.6. LMO2

LMO2 is a crucial element in the axis of LMO2-Prdm16 (PR domain containing 16) involved in the development of hematopoietic stem cells through the endothelial-to-hematopoietic transition as a form of endothelial plasticity [244]. Moreover, in the zebrafish, LMO 2 is a main fate plasticity messenger that determines the decision between endothelial and pronephros tubule cells [245].

#### 4.1.7. ETS (ETV2)

ETS2 plays a pivotal role in endothelial plasticity in the context of specification and early differentiation from mesodermal progenitors [246]. This transcription factor has several lines of action, participates in the migration of progenitor ECs [247], and regulates endothelial reprogramming [229,248].

ETV2 is able to directly reprogram cells (shown on fibroblasts) into endothelial fate lines [249] and to upregulate cell migration through increasing gene chromatin accessibility, regulating cell migration [247].

#### 4.1.8. TWIST-1

The role of this transcriptor factor is strongly associated with epithelial-to-mesenchymal transition (EMT) and is well studied, mainly in tumorigenesis [250,251]. Endothelial cells actively participate in tumor neovascularization and exhibit high levels of plasticity and heterogeneity during this process. TWIST1 stimulates EMT by representing E-cadherin expression and upregulating mesenchymal markers such as vimentin [250,251].

#### 4.1.9. HIF1

As a universal protective molecule against low oxygen concentrations, HIF1 is involved in the induction of VEGF [252] and the modification of gene expression that promotes angiogenesis and vascular remodeling [253]. Additionally, HIF1 is potentially linked with EndMT and may influence cancer and other pathological conditions [253].

To summarize, the concept of endothelial plasticity is not only an ability of ECs to adapt their own initial genetically determined programs to the local epigenetic landscape, but an ability to induce plasticity of surrounding cells belonging to different cell types. ECs are able to build an endothelial-friendly local environment [254,255,256,257], e.g., co-culturing ECs with tanycytes induces cytoskeleton remodeling in the latter through the NO-mediated signaling pathway with cGMP and involvement of prostaglandins and thromboxanes [258]. Herein, ECs indirectly control the glial–neuronal interactions in the brain. To summarize, the concept of endothelial plasticity is not only the ability of ECs to adapt their own initial genetically determined programs to the local epigenetic landscape, but the ability to induce plasticity of surrounding cells belonging to different cell types. ECs are able to build an endothelial-friendly local environment [242,243,244,245], e.g., co-culturing ECs with tanycytes induces cytoskeleton remodeling in the latter through the NO-mediated signaling pathway with cGMP and involvement of prostaglandins and thromboxanes [246]. Herein, ECs indirectly control the glial–neuronal interactions in the brain.

## 5. Endothelial Diversity along the Vascular Bed

### 5.1. Diversity of Ecs in the Microcirculation

ECs at the microcirculation level (arterioles, capillaries, post-capillary venules with a lumen diameter of less than 100 μm) demonstrate more substantial heterogeneity than the macrovessels [259,260,261]. Arterial, venous, and lymphatic micro/macrovessel ECs across the same tissue have similar morphology and demonstrate analogous transcriptomes. Despite standard endothelial features, the transcriptomes in different tissues are also entirely different [262]. *Local epigenetic factors* that influence microvascular diversity are shear stress, intravascular pressure, and metabolic clues [263]. *Genetic influence* is essential for the biochemical organization of the endothelial basal lamina, a thin sheet of extracellular matrix (ECM) located at the epithelial cells’ basal surface. It can be visualized by light microscopy and consists of a lamina lucida or lamina rara (composed of laminin, integrins, entactins, dystroglycan), a lamina densa (collagen IV type, perlecan), and a lamina fibroreticularis. The lamina lucida and lamina densa are named in the literature as the basal laminae and can be visualized only through electron microscopy [264,265,266]. The specific microvascular gene expression patterns are primarily implicated in producing particular basal lamina proteins, including laminin, collagen (4a1 and 4a2), and ECM-interacting proteins (cD36, α1 integrin, α1 integrin, β4 integrin). The microcirculation-specific genes regulate the individual biochemical characteristics of the endothelial cell membrane. Microcirculatory ECs have a surface covered with a negatively charged glycocalyx (composed of glycoproteins and glycolipids). Glycocalyx thickness varies between different parts of the microcirculatory tree [267,268]. The ECs of the fenestrated glomerulus have a thick and specialized glycocalyx (e.g., playing a role in the filtration barrier), whereas the ECs of the fenestrated sinusoidal capillaries of the liver have a thin glycocalyx layer, allowing for quick exchange between the blood and space of Disse [269].

At the microcirculatory level, endothelial morphology is strongly correlated with its functions. ECs are divided into several phenotypes: (1) continuous (have continuous basal lamina and have no fenestrations) (Figure 2) and (2) fenestrated (have fenestrations), with several subtypes (Figure 3 and Figure 4): (a) pseudo-fenestrated (fenestrations covered by fenestral diaphragm), (b) glomerular fenestrated (without fenestral diaphragm); and (c) discontinuous fenestrated cells (lack of basal lamina) (Table 3, Figure 3) [26,270,271,272,273,274,275,276,277,278,279,280,281,282,283,284,285,286,287,288,289,290,291,292,293,294,295,296,297,298,299,300,301,302,303].

**Table 1 cells-13-01276-t001:** Regulating factors of vascular development.

Vasculogenesis Differentiation of Endothelial Precursor Cells (EPCs) into ECs and De Novo Formation of the Primitive Vascular Network (ECs from Mesoderm)		
Phases:	Signaling and transcriptional regulators:	Markers of endothelial differentiation:
I. Extraembryonic vasculogenesis: starts in ~ mice E6.5 embryos, ~3 weeks in humans (yolk sac, allantois, placenta) [59,61,65]	1. Fibroblast growth factor (FGF) family includes 18 paracrine and endocrine factors [56,62,69,73,74] 2. The hedgehog family: Shh, Ihh, Dhh [70] 3. Vascular endothelial growth factor (VEGF) and vascular endothelial growth factor receptors (VEGFRs): VEGFR-2 and VEGFR-1 [56,57,64,67,68,69,74,75,76,84] 4. Neuropilin1(NRP-1) and Neuropilin2 (NRP2) [63,65,67,68,72,76] 5. Transforming growth factor β(TGFβ) and transforming growth factor receptors (TGFRs) [56,57,63,69,70,74,75] 6. Angiopoietins 1 and 2 (ANG1 and ANG2) binds tyrosine kinase with immunoglobulin-like and EGF-like domains 1 and 2 (Tie1 and Tie2) and affect the remodeling of capillary plexuses [70] 7. Platelet-derived growth factor (PDGF) recruitment of pericytes and smooth muscle cells [57,63,65,70,72,74,76] 8. GATA proteins [61,65,67] 9 Krüppel-like factors [61,63] 10. ETS proteins regulate ECs differentiation [61] 11. Homeodomain proteins (HOXB3 and HOXD3) participate in morphogenesis of vascular tube formation [62,70,90] 12. Epidermal growth factor like domain 7(EGFL7) participation in separation and arrangement of angioblast) [59,61] 13. Fibronectin and its receptor α5β1 [65] 14. SOXF factors: SOX7, SOX17, and SOX18 [66] 15. Granulocyte colony-stimulating factor (G-CSF) attenuated delayed tPA [63,71] 16. Overexpressed hypoxia- inducible factor (HIF-1α) stimulates endothelial progenitor cells [64,76] 17. Increase in intracellular Ca^2+^ concentration [68] 18. Bone morphogenetic protein (BMP) signaling pathway [63] 19. retinoid acid [67] 20. Wnt β-catenin signaling [56,61,67] 21. T- box transcription factor gene 18 (TBX18) [56] 22. Wilms tumor transcription factor 1) WT1 [56]	Vascular endothelium cadherin VE-cadherin [59,61,63,65,67,71,76] von Willebrand Factor (vWF) [62] CD34 (early angioblasts and endothelial progenitors) [59,62,63,71,72,73] T-cell acute lymphocytic leukemia (TAL1) [61,65,84] platelet endothelial cell adhesion (PECAM-1)/CD31 [56,59,61,63,73] Tyrosine kinase with immunoglobulin-like and EGF-like domains (Tie-1) and tyrosine kinase with immunoglobulin-like and EGF-like domains (Tie-2) [61,63,65] Flk1 and Flt1 [61,65] Thrombospondin type1 domain-containing protein 1 (THSD1) [62]
A—Assemblation of blood island within the mesodermal layer (yolk sac) [61]		
B—Hemangioblast formation from blood islands (hemangioblast inner part gives rise to the hematopoietic precursors, the outer part give rise to angioblast, differentiation in situ) [61]		
C—Primitive extraembryonic vascular (may contain primitive erythrocytes) plexus organization and ECs differentiation happens in association with hematopoietic precursors in blood islands [61,65]		
II. Intraembryonic vasculogenesis: It starts in ~ mice E7.3 embryos and gives rise to the endocardium great vessels and is not associated with blood formation. There are two types of intra-embryonic hemangioblast forming vascular plexus (usually without erythrocytes) [61,65,73]		
A—Hemangioblast from splanchnopleuric mesoderm (visceral) associated with the endoderm (production of hematopoietic cells and paraaortic splanchnopleura) [63,69]		
AA—Hemangioblast from somatopleuric mesoderm (parietal) associated with ectoderm (give rise to all types of cells, except hematopoietic stem cells) [63,69,73]		
B—Primitive intraembryonic vascular plexuses formation: ECs differentiate from mesoderm as solitary angioblast without the concomitant differentiation of hematopoietic stem cells, except for small regions in the aorta (paraaortic clusters) [61]		
III. Functional vascular network formation in vasculogenesis: primary capillary extraembryonic plexuses anastomose with intraembryonic vasculature through the vitelline arteries and veins and then connect with developing heart tube [61]		
**Angiogenesis** Growth of primary and secondary vascular plexus from pre-existing blood vessels (vessels which lack a fully developed tunica media) in prenatal or postnatal life		
Mechanism and phases:	Signaling and transcriptional regulators:	Markers of endothelial differentiation:
**I. Sprouting mechanism**—based on endothelial cell migration, proliferation, and tube formation with or without blood flow (the result is new vascular tube formation) [56,64,67,68,69,73,84]	1. Vascular endothelial growth factor VEGF [56,57,61,62,63,65,67,69,70,71,72,74,75] 2. Transforming growth factor β,α (TGFβ,α) [64,66,70,71] 3. Angiopoietins and tyrosine kinase with immunoglobulin-like and EGF-like domain receptors (Tie receptors) [65,70,74] 4. Platelet-derived growth factor (PDGF) and Platelet-derived growth factor receptor-β (PDGFR-β) [65,73] 5. SOXF factors: SOX7 and SOX18 [66] 6. Atypical chemokine receptor CXCR7(ACKR3) [76] 7. Angiopoietins 1 and 2 (ANG1 and ANG 2) [69,70,74] 8. Granulocyte colony-stimulating factor (G-CSF) attenuated delayed tPA [63,71] 9. hypoxia-inducible factor HIF-1α [65,71] 10. Increase in intracellular Ca^2+^ concentration [68] 11. High mobility group box 1 (HMGB-1) [60] 12. Connective tissue growth factor (CCN2) [60] 13. Delta/jagged-NOTCH signaling [61,63,65] 14. Metalloproteinases MMP [63,69] 15. Ephrin- B (EPH-B) [63,69] 16. Semaphorins (SEMA 3 proteins) [63,65,70] 17. Rho-associated protein kinase (ROCK) [63] 18. Chicken ovalbumin upstream promoter transcription factor (Coup-TFII) [84]	Vascular endothelium cadherin (VE-cadherin) [59,61,63,65,67,71,76] von Willebrand Factor (vWF) [62] CD34 [59,62,63,71,72,73] Platelet endothelial cell adhesion (PECAM-1)/CD31 [56,59,61,63,73] Endoglin/CD105/ [65,72] THSD1 Thrombospondin type1 domain-containing protein (THSD1) 1 [62] Krüppel-like factor 4(KLF4) [63] A disintegrin-like and metalloprotease with thrombospondin type 1 repeats 13 (ADAMTS-18) [62] Aminopeptidase N APN(CD13) in tumorigenesis [72] Intercellular adhesion molecule-1(ICAM-1 or CD54) [72]
A. **Neovessel growth**—the disintegration of the basal lamina of existing vessel, migration and proliferation of ECs, lumen formation, and loops organization by sprouts and anastomoses [226,239]		
B. **Neovessel stabilization**—delay of the endothelial proliferation, basal lamina reconstruction, coverage of the immature vessel with pericytes [63,67]		
**II. Intussusceptive microvascular growth (IMG) mechanism:** based on the division of existing vessel lumens by formation and insertion of tissue folds and interstitial cellular columns into the lumen of pre-existing vessels (lumen expansion occurs through the organization of new units of extracellular matrix). Blood flow-dependent process [63,69]		
**A. Interendothelial “transluminal bridge” formation:** ECs located at the opposite side of the capillary wall move near to each [63]		
**B. “Cylinder tissue bridge” establishment:** tissue form as a cylinder bridge, which perforates endothelial bilayer and extends across the lumen. Then ECs cover the cylinder tissue bridge involving the cytoplasmic extensions of myofibroblast and their microfilaments inside the cylinder’s core [63]		
**C.” Pillar” formation:** the framing of the pillar by pericytes close to the lateral part wall [63]		
**D.” Pillar” growth:** pillar growth into an intercapillary mesh [63]		

*Continuous ECs* are present in tissues with low but well-controlled exchange. This endothelial cell type’s primary characterization is the continuous basal lamina and lack of fenestra (Figure 2, Table 3) [304]. In contrast, the fenestrated and sinusoidal endothelium is mainly located in the organs with high uni- or bidirectional substrate diffusion.

*Pseudo-fenestrated ECs* are located in endocrine tissues, gastrointestinal mucosa, and renal peritubular capillaries. They have a continuous basal lamina and pores (60–70 nm in diameter), with a thin diaphragm (Figure 3, Table 3).

*Discontinuous fenestrated* ECs (or sinusoidal ECs) are characterized by a lack of a continuous basal lamina, multiple fenestrations with 50–100 nm in diameter aggregated into a group of 10–100 fenestrae (liver “sieve plates”), and big pores (100–200 nm in diameter) without a diaphragm. These types of ECs are typical for the liver, spleen, and BM (Figure 4, Table 3).

Glomerular fenestrated ECs take the intermediate position between common pseudo-fenestrated and discontinuous fenestrated ECs: they have a basal lamina, pores with diameters of 60–80, and no diaphragms (Table 3) [293,297,305]. The local microcirculatory functions and morphology contribute to permeability regulation, i.e., an exchange between the vascular lumen and the extravascular tissue [263,306,307]. The above-mentioned phenotypic differences of ECs directly affect microcirculatory endothelial permeability [308].

*High endothelial venules (HEVs)* are a special part of the microcirculatory bed [309]. These vessels belong to post-capillary venules, which mostly accompany lymphoid tissues such as lymph nodes, Peyer’s patches, and tonsils. The specificity of these cells is strongly related to immune system function and serves to facilitate the entry of lymphocytes from the bloodstream into the lymphoid tissues (place of immune response initiation and regulation). The ECs of HEVs have a flat and cuboidal shape, unlike the flat and elongated morphology of typical ECs. Notably, the surfaces of these types of ECs display a specific glycosylation pattern composed of Pereferial node addressins (PNAd) and express high levels of sialomucins, which contribute to lymphocyte homing through binding L-selectin to lymphocytes [310]. The listed structural and functional characteristics facilitate the mediation of lymphocyte trafficking into lymphoid tissues, supporting effective immune surveillance and response [311].

Critical structural elements of the endothelia, regulating the cellular diffusion process (especially across the continuous endothelium) and layer stability, are cell-to-cell junctions [312]. Junctional endothelial diversity is another marker of EC heterogeneity, characterized by variations in the expression of junctional proteins [313]. The junctions connect to cytoskeletal proteins and several signaling proteins that contribute to the maintenance of shape and polarity [314,315]. There are three types of cellular junctions: tight junctions (TJs), adherence junctions (AJs), and gap junctions (GJs) [272].

TJs are associated with barrier tissues, where the fast interchange between the blood and tissues does not exist or is strictly regulated. These types of junctions are typical of the endothelium in the brain, lungs, endocardium, nerve capillaries, fat, and muscle tissue. They display the role of a selective wall, allowing for the select paracellular passage of ions between cells [316,317]. The claudin family (26 proteins in humans), occludin, and tricellulin are responsible for the TJs’ barrier specificity [318,319]. In these tissues, TJs are not only responsible for the barrier function (blood–brain barrier, air–blood barrier, and blood–nerve barrier), but also for the physical stabilization of macrostructures with regular physical stress, e.g., muscle and endocardial ECs adapt quickly to shape changing during muscle contraction [320,321,322,323,324].

AJs are cell–cell adhesion complexes interacting with the F-actin cytoskeleton and contribute to tissue homeostasis, embryogenesis, stabilization, initiation of cell–cell adhesion, and control of intracellular signaling. These multiprotein complexes are composed of cadherins and nectins [325]. TJs and AJs are zipper-like elements localized at the lateral membrane of the ECs [326]. The function and stability of EC AJs are regulated by various types of interactions with actin cytoskeleton participating in modulation of endothelial permeability as a reaction to different types of stimuli [327].

GJs are members of the connexin protein family which form intercellular channels and provide direct communication patterns between endothelial and surrounding cells: electrical coupling and flow of metabolites in exchange [328]. GJs proteins are predominantly tightly associated with the plasma membrane and, in addition to being passively transported, may also play a role in the rough endoplasmic reticulum, mitochondria, and Golgi apparatus [329]. GJs participate in transmitting vasodilatory signals from the capillary network to arterioles, and they conduct signals from the endothelial to the muscle cell layer in some vascular beds [330,331]. They contribute to the chronic remodeling of vessels through the induction of cellular stiffness, actin rearrangement, and activation of pro-inflammatory genes that result in disease development [332,333]. GJs are mainly located in organs with a rapid molecular exchange between the blood and the surrounding compartment (endocrine glands, lymphatic capillaries, liver sinusoids, spleen, and BM). GJs facilitate fast information exchange that helps the whole endothelium to respond to a focal signal in a coupled fashion, like a syncitium [62,334,335,336].

### 5.2. Diversity of Ecs in the Large Blood Vessels

The expression of genes specific for the macrocirculation bed contributes to modulation and biosynthesis of the corresponding ECM: fibronectin, collagen (5α1 and 5α), and osteonectin [337]. The main epigenetic microcirculatory factors are arterial pressure, hypercholesterolemia, and inflammation [338,339,340].

Morpho-functionally, arterial ECs (AECs) and venous ECs (VenECs) specialize in the non-stop conduction of oxygen-rich/poor blood towards or away from the heart. Macrocirculatory ECs participate in the systemic control of blood pressure and the regulation of blood flow (shear stress) and vasomotor tone (circumferential wall stress). Therefore, they are also involved in the maintenance of the total peripheral vascular resistance (elastance and compliance) and vascular capacitance, activation and migration of blood cells/immune cells, as well as in vascular disease development [263,306,307]. AECs have elongated and narrow shapes. They contribute to the tunica intima, which is reinforced by smooth muscle cells. In comparison with VenECs, AECs do not participate in the formation of valves. Additionally, AECs provide fewer conduits for transmigration of immune cells [341,342] and play a lesser role in terms of inflammation when compared with the venous endothelium [343]. Arterial and venous ECs belong to the continuous endothelial cell type. AECs and VenECs contain TJs and AJs, which provide cell–cell interactions and secure vessel stability (providing laminar blood flow and reducing vascular shear stress).

### 5.3. Lymphatic Macro/Microcirculation

The lymphatic system belongs to the drainage system of the body, and is closely related to blood circulation [344]. It furnishes the peripheral immune system [345] and nervous system with drainage of the extracellular liquid and corresponding clearance of antigens contributing to the maintenance of brain homeostasis [346,347]. The lymphatic system’s essential functions include the removal of interstitial fluid and the maintenance of local homeostasis; local tissue immunological supervision; and host defense, which entails cell trafficking and transcytotic delivery in the guidance and support of immunocompetent cells. It is also involved in the absorption of dietary lipids and participates in vessel and organo-vascular morphogenesis [348]. The morpho-physiological organization of the lymphatic bed determines its functionality.

The lymphatic capillaries consist of a single layer of lymphatic ECs (LECs) characterized by different cellular integrity levels and the absence of a basal lamina, in contrast to most blood vessels [349]. The “initial blind-ended” capillaries eliminate intercellular overflow and regulate the macromolecular balance in the interstitial space. LECs in the initial capillaries interconnect directly with the interstitial matrix, forming the specific discontinuous oak-leaf shape and button-like junctions (buttons) [350]. The larger collecting capillaries drain into the thoracic or right lymphatic duct and, finally, to the brachiocephalic veins. The level of cell–cell integrity between “collecting” LECs is higher as compared to the “initial” LECs, which closely cooperate with smooth muscle cells or pericytes, build continuous zipper-like junctions (zippers), and additionally acquire intraluminal valves. This architecture provides permanent unilateral lymphatic flow [351]. Ontogenetically, the zippers are older than the buttons [352]. Both junction types demonstrate a high capacity for plasticity under various physiological and/or pathophysiological demands; mature junctions are able to reorganize junctional proteins and the cytoskeleton [283]. Inflammation and infection stimulate the transformation of the existing button-like junctions into zipper junctions [302]. Organ-specific functions may be determined by the source of lymphatic origin and cellular microenvironment [348]. In early development, the lymphatic vessels arise mainly from large veins. In the postnatal period, lymphatic development occurs by reorganizing lymphatic capillaries or by transdifferentiation of venous, mesenteric, and hemogenic endothelium [353]. Notably, human and mouse myeloid lineage cells can successfully transdifferentiate to LECs through the activation of toll-like receptor-4 (TLR4) [354]. PROX1, SOX18, NOTCH, Wingless-related integration site (WNT), COUP transcription factor 2, AM-CRL-RAMP2, angiopoietin-TIE, VEGF, VEGFR3, and sphingosine 1-phosphate (S1P)- sphingosine 1-phosphate receptor (S1PR1) participate in the control of the lymphatic fate [355,356,357,358,359]. VEGFR3, VEGFC, PROX1, NOTCH, and LYVE l are responsible for lymphatic migration and sprouting; the specification of LECs is derived by WNT, BMP, and JAGGED1/NOTCH1 signaling pathways [360]. The angiopoietin-TIE is necessary for cardiovascular and lymphatic development, including remodeling [361,362]. S1P and S1PR1 also participate in LEC remodeling, development, lymphatic valve formation, endothelial barrier function, dilatation of vessels, and inflammation [363]. The dysfunction of the lymphatic system may result in the initiation and progression of a wide range of diseases. Decreased lymphatic drainage and cerebrospinal fluid flow aggravate glioblastoma progression and may influence the onset of pre-senility in the elderly [347,364]. Metabolic disorders such as diabetes mellitus, obesity, and metabolic syndrome are closely related to the dysfunction of the lymphatic system. They are characterized by chronic inflammation, low inflammatory answer, development of secondary lymphedema, and lipedema [365,366].

## 6. Selected Tissue-Specific Endothelial Phenotypes

The brain, heart, lungs, liver, kidneys, gut, and endocrine organs have distinct endothelial subpopulations displaying specific characteristics.

*Brain endothelial cells (BECs),* along with pericytes and astrocytes, form the blood–brain barrier and enforce the protective barrier function [367]. The relationships between pericytes and ECs (including BECs) are dynamic and vital for the integrity and maintenance of vessel walls [368]. Functionally, pericytes play a vital role in modulating blood flow, supporting angiogenesis, and contributing to the stability and maturation of blood vessels [369]. These functions, first of all, are related to specific morphological organization and, secondly, to metabolic interactions between these types of cells. Morphologically, pericytes are embedded in the basal lamina of the ECs (including BECs) within capillaries and postcapillary venules [370,371,372]. This anatomical proximity facilitates direct cell–cell communication, mainly through the Gap junctions, positioning pericytes as principal homeostasis regulators of the local microenvironment and vessel stability [370]. Metabolic interaction between ECs (including BECs) and pericytes, based on pericytes’ preferential use of oxidative phosphorylation (OXPHOS) for ATP production, makes pericytes the primary stabilizers of vessel wall integrity in the vascular system [373]. Through these diverse functions, pericytes ensure the proper functioning of the vascular system and respond to various physiological and pathological conditions over the vascular bed.

BEC morphology itself is specific and characterized by numerous caveolae, endothelial TJs, AJs, and an increased number of mitochondria [374]. The BEC surface displays dozens of specific proteins, regulating cell–cell communication and molecular transport: claudin-5, occludin 3,12 (OCCLN 3,12), endothelial cell-selective adhesion molecule (ESAM), junction molecules 1, 3 (JAM1,3), tricellulin- and lipolysis-stimulated lipoprotein receptor (LSR), vascular endothelium cadherin (VE-cadherin), N-cadherin, and transporters (Glut-1, Slc2a1) [375,376]. Notably, the BECs do not express thrombomodulin [377]. The morphology, proteome composition, and expression of CD31 and von Willebrand factor differ in various brain parts and types of vessels (capillaries, arterioles, venules) [378,379]. The local cellular environment mediates BEC functionality: microglia change BBB (blood–brain barrier) permeability, also causing systemic inflammation [380], and BEC–macrophage communication instigates barrier dysfunction in patients with hypertension [381]. Plasmodium parasites initiate BBB disruption, resulting in edema [382]. Gamma interferon contributes to the BBB leakage process [383].

*Endocardial ECs (EECs)* display characteristics typical of the continuous vascular endothelium. The EECs form a thin cell layer with varying thicknesses ranging from 50 to 300 μm. Primarily, they contribute to normal blood flow and physiological cardiac function. The total amount of EECs in the heart corresponds to 2–3% of the heart mass [384]. These cells are characterized by the unique presence of numerous microvilli on the surface, and their basal lamina is composed of delicate collagen and elastic fibers [385]. Compared to vascular ECs of arteries, veins, and capillaries, the EECs have a broader shape; particular cellular connections (e.g., GJs) and intercellular spaces define the specificity of these EECs. The GJs control the transendothelial permeability of EECs through quick passage of charged ions (mostly Ca^2+^), messenger molecules, and small metabolites [386]. Metabolically active EECs use fatty acid and lactate as energy sources [387] and control long-chain fatty acid (LCFA) delivery and metabolite transport to myocytes [388]. GLUT-1,3,4 mediates the primary glucose uptake in EECs, while GTPases participate in cellular protein transport [1,389]. The condition of EECs is regulated by systemic and local NO synthases, endothelin-1 (ET-1), angiotensin II, prostacyclin, natriuretic peptides A and B, VEGF, hepatocyte growth factor (HGF), FGF, interleukin-6 (IL-6), thrombospondins, and insulin-like growth factor-1 (IGF-1) [390,391]. EECs express various markers that are typical for all endothelial cells, including CD31, CD34, vWf, caveolin, neuregulin-1 (NRG-1), and VE-cadherin [392]. The heart endothelium can be considered as an active paracrine, endocrine, and autocrine organ that produces cardioprotective substances like secretory leukocyte protease inhibitor (rhSLP1) [393]. Simultaneously, EECs firmly contact the cardiomyocyte surface (between the two types of cells, there is a thick fibrillary basal lamina). This cell–cell cross-talk is essential for heart regeneration and remodeling [390,394]. Release and diffusion of signaling molecules within this space participates in heart inotropy [395]. However, any extrinsic cues (factors located out of the cardiovascular system) provoke disbalance in endothelial physiology, e.g., hyperthyroidism is regarded as a factor of cardio-cerebrovascular dysfunction [396], and autoimmune diseases as well as systemic and local inflammation can damage the morphology or physiology of vascular ECs and EECs [397]. Hypoxia may cause upregulation of vWf in EECs through high-mobility group box-1 (HMGB1) and activation of toll-like receptor-2 (TLR2), resulting in an augmentation of the plasma sodium concentration, an increase in E selectin and P selectin, downregulation of anti-thrombotic factors [398], and altered innate immunity reactions [399,400].

*The pulmonary ECs (PECs)* contribute to the gas exchange with the external microenvironment, formation of the air–blood barrier in the alveoli, maintenance of pulmonary and systemic vascular homeostasis, and immune response (through mitochondrial activation of innate immune mechanisms that stimulate lymphatic delivery to the lymphatic nodes and promotion of adaptive immunity). They also provide a failsafe mechanism to balance blood pressure in the lungs and, possibly, to regulate coagulation with PGI2 prostaglandin [401,402]. PECs are classified as vascular (macrovascular and microvascular) or alveolar ECs. Recently, Car^4+^ high PECs (with high levels of Car4 and CD34 expression) were identified by scRNA-seq analysis. This cell subpopulation is located throughout the lung periphery and is primed to respond to VEGFA signaling. The number of Car^4+^ high PECs increases in the regenerating areas of pulmonary tissues after influenza infection [403]. The alveolar endothelium is divided into two intermingled cell types: aerocytes (specialized in the gas exchange and leukocyte trafficking) and general capillary cells (functioning as stem/progenitor cells and responsible for regulating the vasomotor tone of capillaries) [404]. The pulmonary endothelial barrier plays a vital role in vascular homeostasis maintenance. TJs of PECs are formed by occludins, claudins, and junctional adhesion molecules. Notably, primarily vascular endothelial cadherin comprises the AJs [405]. PECS is a continuous type of endothelial cell with an epithelioid shape. When metabolically active, these cells produce prostacyclin, bradykinin, angiotensin, endothelin-1, prothrombotic and anti-thrombotic factors, and other anti-inflammatory cytokines [406,407]. PECS express vWf, endothelial NO synthase, cadherins, CD31, and angiotensin I-converting enzyme (ACEI) [408,409,410].

*The endothelial population of the kidney ECs (KECs*) is diverse. KECs include glomerular ECs (GECs) and peritubular capillary ECs (PCECs). Both types of cells have a continuous basal lamina; however, PCECs belong to the pseudo-fenestrated and GECs to the true-fenestrated ECs (Table 1) [411]. Pseudo-fenestrations of PCECs represent incomplete fenestration with an overlying fenestral diaphragm, controlling molecular passage and exerting a “sieving” function or regulating the counter-current-based gradient in the medulla and cortex [412]. Fenestral diaphragms are typical for PCECs and intestinal ECs (but not for GECs and form open holes); they are composed of radial heparan sulfate proteoglycan fibrils (glycocalyx tufts) acting as a permselective barrier and regulating the passage of water and small molecules [296]. Electron microscopy of GECs has revealed a thick (around 200 nm) glycocalyx layer [411,413,414]. GECs express vWf, vascular cell adhesion protein 1 (VCAM1), and intercellular adhesion molecule-1 (ICAM-1). GECs, together with mesenchymal cells, produce HGF, Kruppel-like factor (KLF), and insulin-like growth factor-binding proteins (IGFBPs) through the activation of the c-Met receptor [415]. PCECs sit on the continuous basal lamina, which separates them from the pericytes. PCECs display some immunohistochemical characteristics of macrophages: OKM5 (medullary expression only, there is no expression of OKM5 in GECs) and interleukin-1 (IL-1) expression [416,417]. In adults, PCECs display CD31 and VE-cadherin, a low level of vWf, and overexpression of the plasmalemma vesicle-associated protein 1 (PV1). GECs do not have the PV1 [286]. According to recent data, CD34 has been revealed in the peritubular microvasculature of the adult human kidney, and the level of expression usually correlates with the severity of glomerular and tubulointerstitial damage [286,418,419].

*Intestinal ECs (IECs)* belong histologically to the pseudo-fenestrated endothelium with TJs and fenestral diaphragms. These cells are essential for local and general immune responses and the development of intestinal inflammation [420]. The IECs possess fenestrated diaphragms. This fenestrated diaphragm, along with the gastrointestinal mucosa, forms a selective barrier between the extracellular and intravascular space (so-called gut-blood barrier) [421]. The healthy intestinal barrier is impermeable to 70 kDa molecules [422]. TJs and claudin proteins regulate the functionality of the intestinal microvascular endothelial barrier. The overexpression of claudin-1 increases the antiviral and antibacterial resistance of IECs by intensifying the mucosal and endothelial integrity [423]. Downregulation of claudin-5 and claudin-8 increases barrier permeability [424]. The tight junctions belong to a dynamic structure that can adapt its protein composition according to external (pathological or physiological) stimuli [425]. The primary markers of IECs are CD31, vWf, VE-cadherin [426], and E-selectin. P-selectin, VCAM-1, and ICAM-1 cannot be detected in the basal, unstimulated state. However, the pro-inflammatory cytokines Il-1β and tumor necrosis factor-α (TNF-α) can induce the biosynthesis of the latter proteins [427,428].

*The liver sinusoid ECs (LSECs)* are metabolically active, organ-specific ECs with great transdifferentiation potential. Phenotypically, LSECs form a discontinuous fenestrated endothelium with a lacking basal lamina [429]. LSECs are organized in sieve plates and display plenty of fenestrations: ~2–20 fenestrations per μm^2^, corresponding to 2–10% of the LSECs surface. The diameter and distribution of fenestrations allow for rapid exchange between the space of Disse and the blood [430]. LECs are essential for: (1) the primary selective barrier (protecting liver parenchyma); (2) the formation of scavenger and endocytosis systems of the liver; (3) immune response; (4) paracrine signaling; and (5) liver regeneration [431,432]. Discontinuous fenestrated LECs express such specific markers as stabilin-1, stabilin-2, liver endothelial differentiation association protein (LEDA-1), and CD32b [433]. Additionally, mannose receptors (CD206) and toll-like receptors are present on the surface of the LSEC, while the CD31 and vWf are not expressed in the LSCs under normal conditions or in young individuals, but may be present in ECs of larger blood vessels and lymphatic vessels in the liver [434,435]. VEGFR3 is often used as a specific marker for LECs (FLT-4) [436].

*Splenic sinusoidal ECs (SSECs)* are the most abundant non-immune cells of the spleen which participate in forming the splenic sinusoidal wall and create a discontinuous endothelial layer adjacent to the fenestrated basal membrane. The spleen is a secondary lymphatic organ involved in immunological supervision, clearance of the blood, and maturation of immune cells [437]. The endothelium forms filamentous structures, contributing to radial construction and free, yet limited and retarded, blood passage [438,439]. Open blood flow is found in splenic cords [440]. The fibers around the SSECs are associated with cellular VE-cadherin, β-catenin, p120 catenin, and actin filaments [441]. Thus, SSECs, together with spleen littoral cells, specialize in the filtration of senescent red blood cells [442]. SSECs are characterized by expression of CD31, CD8 α/α, CD271, stabilin-1, and CD206 [442,443]. The vWf is another marker of SSECs which is upregulated by hypothermic conditions [444].

## 7. Alteration of Endothelial Cells and Pathologies

### 7.1. Cancer

ECs are structural units of the circulatory system which are essential for cancer development [445]. Cancer onset and progression are associated with the metabolic reprogramming of ECs and the formation of specific cancer-hospitable “niches” or “sub-niches” that enable tumor-associated AG and facilitate tumor growth in the body [446]. Tumor progression is an active metabolic process requiring persisting neovascularization through VG, AG, LyAG, intussusceptive AG, vessel co-option, and vasculogenic mimicry [447]. VG in tumor development is characterized by the recruitment of endothelial progenitor cells, contributing substantially to tumor formation. Similarly to AG in other contexts, cancer-related AG involves the formation of vessels from mature ECs through a sprouting process [448]. Intussusceptive AG constitutes a splitting process of an initial vessel lumen through invagination, thereby creating a new vessel [449]. The usage and alteration of existing vessels for cancer cell spreading is called “co-option” [450]. Vasculogenic mimicry in tumorigenesis is the formation of vasculogenic-like channels without the EC layer [LIT]. This is necessary for the direct transport of the tumor cells in the vascular bed [451]. Notably, the amount of ECs and precursor blood cells in patients with different types of cancer (cell lung cancer, ovarian cancer, leukemia, myeloid leukemia, hepatocellular carcinoma, and breast carcinoma) is substantially increased [452], since the cells are more often released from the BM or the systemic vascular wall [453,454]. Different populations and subpopulations of ECs, cancer-associated fibroblasts, and tumor-associated immune cells (especially myeloid-derived innate cells) can be reprogrammed by the tumor environment and incorporated into the tumor stroma [455]. Tumor progression and neovascularization require the transformation of ECs and stromal cells into morphologically abnormal, aneuploid tumor-derived ECs (TECs). TEC lines in tumor vessels display loosened junctions and increased vascular permeability, leading to tumor dissemination and the formation of hemorrhages [456]. Membrane storage in transformed endothelial cells may also contribute to the formation of larger vessels in the form of VVOs (vesiculo-vacuolar organelles) that add to the rapid extension of the bilayer surface area [457]. Cancer-related EC reprogramming results from metabolic re-education and morphological transformation, e.g., through EMT/EndMT [458]. Tumor EMT/EndMT is a dysregulated process compared to EMT/EndMT in embryonic development and tissue regeneration [459]. This process is induced and controlled by the following conditions and factors: hypoxia [460], TGF-β [461], survivin [462], endosulfan [463], and gremlin-1 [45,464]. It has recently been demonstrated that ECs promote metastatic cancer dissemination through differential expression of adhesion molecules (E-selectin, P-selectin, integrin, transpanin, trombospodins), secretion of CCL5, and enhancing autophagy [465,466]. ECs in healthy body organs are also programmed by the transfer of tumor molecules from the environment and can aid in the metastasis of the original tumor tissue [467].

**Table 3 cells-13-01276-t003:** Morpho-functional classification of vascular ECs.

Morpho-Functional Characteristics	ECs Types				
	Continuous	Fenestrated Endothelium			Lymphatic
		Pseudo-Fenestrated Fenestrated	Glomerular Fenestrated Endothelium (True Fenestrated)	Disconntimous Fenestrated (Sinusoid) Endothelium	
Localization	Brain [1,277,278] Skin [1,270] Lungs [1,295,296,302] Heart [1,270,293,295,299,302] Arteries [1,270,272,289,292,295,299] Veins [1,270,295,296,299]	Intestinal tube [270,275,278,287,293,296] Adrenal cortex [273,278,293,295] Pancreatic islets [278,293] Kidney peritubular capillaries [278,289,293,296,301]	Kidney (Glomeruli) [270,275,290,293,295,296]	Liver [270,278,293,296,299] Spleen [270,278,293] Bone-marrow [275,278,293,295,299]	Lymphatic vessels and lymph nodes [232,278,283,288,300,302]
Function	Highly selective barrier: transfer of water and small solutes (diameter ~6 nm), transport of big molecules occurs through channels or transcytosis [270,278]	Size and selective charge barrier: permeable for small molecules and water, but impermeable for macromolecules (e.g., albumin, peptide hormones) and blood cells [278,286,293]	Low-selective barrier: permeable for small molecules and water and macromolecules (e.g., albumin) but impermeable for cells from ultrafiltrate [290,293]	Non-selective barrier: permeable for water, macromolecules, and blood cells [291,293,297]	Non-selective barrier of lymphatic capillaries (sinusoid lymphatic ECs): permeable for macromolecules and immune cells (high permeability) Selective barrier, collecting lymphatic vessels demonstrate low permeability [288]
Basal lamina	Yes [270,293]	Yes [293]	Yes [293]	Absent or poorly developed [270,291,293]	Lymphatic capillaries (initial capillaries): highly incomplete perforated basal lamina and discontinuous junctions (buttons) [288,294] Collecting lymphatic vessels: continuous basal lamina, continuous junctions (zippers) [294]
Fenestra, nm	No [1,270,276,293]	60–70 [273,293]	60–100 [293,301]	“Sieve plates”: 50–100 “Gaps”: 100–200 [293,296,297]	No [288]
Fenestral diaphragm	No [270,276]	Yes [1,273,293]	No [296]	No [296]	No [288]
Glycocalyx	Yes [275]	Yes [293,296]	Yes [275,290,301]	Yes [269,377,378]	Yes [303]
Non-specific Markers	CD31: Heart (low expression), skin [277,285] CD34: Heart, skin [292] von Willebrand factor (vWF): Heart (low expression), skin, lungs [277,292] CD62E or E -selectin (inducible) [270,277,292] CD62P or P selectin (inducible) [270,277,292] CD106 or VCAM-1 (inducible) [277] CD54 or ICAM (inducible) [277] Flt-1 or vascular endothelial growth factor receptor 1 (VEGFR1, inducible) [277] KDR/Flk or vascular endothelial growth factor receptor 2 (VEGFR2, inducible) [277] CD144 human [26,280]	CD31 [277,285,292] CD34 [277,285,292] von Willebrand factor (vWF) peritubular ECs (low expression) Fli-1 (nuclear) [277,285,292] CD62E or E -selectin (inducible) [277,285,292] CD62P or P selectin (inducible) [277,285,292] CD106 or VCAM-1 (inducible) [277,285,292] CD54 or ICAM (inducible) [277,285,292] Flt-1 or vascular endothelial growth factor receptor 1 (VEGFR1, inducible) [277,285,292] KDR/Flk or vascular endothelial growth factor receptor 2 (VEGFR2, inducible) [277,285,292] vWf [277,285,292]	CD31 [277,285,292] CD34 [277,285,292] Fli-1(nuclear) [277,285,292] von Willebrand factor (vWF) expression) [277,285,292] CD62E or E -selectin (inducible) [277,285,292] CD62P or P selectin (inducible) [277,285,292] CD106 or VCAM-1 (inducible) [277,285,292] CD54 or ICAM (inducible) [277,285,292] Flt-1 or vascular endothelial growth factor receptor 1 (VEGFR1, inducible) [277,285,292] KDR/Flk or vascular endothelial growth factor receptor 2 (VEGFR2, inducible) [277,285,292]	CD31: Liver, Spleen; Bone marrow [277,292] CD34: Bone marrow [277,292] von Willebrand factor (vWF): liver, spleen [61,78] Fli-1: Liver, spleen, bone marrow [277,292] CD62E or E -selectin (inducible) [277,291] CD62P or P selectin (inducible) [277] CD106 or VCAM-1 (inducible) [277] CD54 or ICAM (inducible) [277] Flt-1 or vascular endothelial growth factor receptor 1 (VEGFR1, inducible) [277] KDR/Flk or vascular endothelial growth factor receptor2 (VEGFR2, inducible) [277]	CD31 [277,285,292] CD34 [278,284,292] Fli-1 [284,292] von Willebrand factor (vWF) [284,292] CD62E or E -selectin (inducible) [277,284] CD62P or P selectin (inducible) [277] CD106 or VCAM-1 (inducible) [277] CD54 or ICAM (inducible) [277] Flt-1 or vascular endothelial growth factor receptor 1 (VEGFR1, inducible) [277] KDR/Flk or vascular endothelial growth factor receptor 2 (VEGFR2, inducible) [277]
Specific markers	Angiotensin-converting enzyme ACE or CD143 (human heart and lungs) [277,278] Thrombomodulin (TM): absent in brain endothelial cells [277,278] Tissue non-specific alkaline phosphatase or TNAP: brain (mouse and human and rat) [276,278] Thrombospondin type 1 domain or THSD1: vessels (mouse and human) [278,281] P-glycoprotein or MDR 1a: brain and lungs (mouse and human) [20,277] CD73/ transferrin receptor: brain (mouse and human) [277] Platelet-derived growth factor receptor: brain (human and mouse) [277,282] Sca-1 (mouse pulmonary ECs) [277] HLA-DR (human, pulmonary ECs) [280] Glut-1: brain (human and mouse) [277]	PV1 (human and mouse peritubular capillary) [293,295,296,299] MAdCAM-1 (venules intestinal) [277] Nephrin (human Pancreatic islet) [280] CD117 (mouse pancreatic islet) [278]	ADAMTS-13 (mouse and human) [298]	CD32b (human liver sinusoidal) [280,336,434] LYVE-1 (mouse liver and spleen sinusoidal) [278,304] PV-1 (mouse spleen sinusoidals) [280,296] Angiotensin-converting enzyme (ACE or CD143) Stabilin 1,2 [434] Liver-endothelial differentiationassociation protein (LEDA-1) [434]	CD90 (human and mouse) [280] Flt-4 or vascular endothelial growth factor receptor 3 (VEGFR3, human and mouse) [277] Desmoplakin [277] Podoplanin or PDPN (human) [278,280] LYVE-1 (human and mouse) [278,280] Prox-1 (human and mouse) [280,294] Clever-1 or Stabilin-1 or FEEL-1 (human and mouse) [231,280,303]
Junctions	Tight junctions/adherence junctions [272]	Tight junctions/gap junctions [272,279]	Tight junctions/gap junctions [272,279]	Gap junctions/tight junctions [272,279,302]	Buttons (discontinuous button-like junctions, with openings at the borders of adjacent lymphatic ECs) enriched with adherents and tight junction proteins [350,353] Zippers (continuous zipper-like junctions without openings at the borders of adjacent lymphatic ECs) enriched with adherent and tight junction proteins [350,353]

### 7.2. Endothelial Turnover, Regeneration, and Repair

The turnover of the ECs in the vascular system includes replacing the old endothelial cells with new endothelial cells. This process reflects the balance between endothelial proliferation and apoptosis rates. Endothelial-to-endothelial turnover takes from 47 days to 23,000 days and increases under the influence of intrinsic or extrinsic factors such as hemodynamic forces, shear stress, pressure, ischemia, certain vascular and non-vascular diseases, and continuous local or systematic endothelial injury [468,469,470,471]. Regulation of the endothelial-to-endothelial turnover is controlled by Rasa 3 (decreasing level of Rasa3 increases cell adhesion), occludin, VE-cadherin, ZO-1 (ensuring maintenance of vascular integrity in vertebrates), junctional adhesion molecule C (JAM-C-1 responsible for the endothelial cell migration), VEGF (induces endothelial leakage through the Src-mediated degradation of VE-cadherin), angiopoietin-1, mitogen-activated protein-4 kinase-4 (MAP4K4), and Il-2 [472,473,474,475,476,477]. The turnover is a neighbor-depending process, requiring extra- and intracellular matrix reorganization and loss of cell–cell contacts. The leading suppliers of cells for EC turnover are blood-circulating endothelial cells, local endothelial progenitor cells, and bone marrow-derived SCs [468,478]. The endothelial turnover in the brain is low. However, the situation changes in the case of brain injury, where an increased amount of the neurovascular unit cells (ECs, pericytes, astrocytes) or endothelial progenitors may participate in endothelial recovery [479]. Compared to other microvascular ECs, the human brain endothelial cells react differently to shear stress factors, i.e., the turnover rate decreases with the increasing shear stress level [480]. According to the “nuclear bomb-derived C14 analysis,” the heart/cardiac endothelial cell turnover is high (more than 15% of all ECs per year) compared with mesenchymal cells (less than 4% per year) or cardiomyocytes (less than 1% per year) [481]. In contrast to the brain ECs, in injury of the heart, the pre-existing ECs are the primary source for endothelial recovery [482]. The daily pulmonary endothelial turnover involves approximately 1% of all ECs [483]. In the case of lung injury, the local EPCs are recruited in the turnover process [484]. The sinusoidal endothelial cell progenitors can replace the mature LSECs and the intrahepatic or resident sinusoidal ECs. Interestingly, those mature LSECs can proliferate and expand with the involvement of VEGF and FGF. BM progenitor cells, on the other hand, do not participate in liver turnover under physiological conditions, but become relevant in liver pathologies [291,485,486]. The splenic endothelial turnover recruits the local EPCs, which closely interact with the sinusoidal endothelial cells [487]. Endothelial derivates make up around 30% of the spleen [488]. Local EPCs (colony-forming and pro-angiogenic cells) are the principal elements of kidney EC turnover [489]. The intestinal EC homeostasis also builds on a unique, local intestinal stem cell niche for renewal and proliferation [490].

Endothelial repair is a vital process for endothelial physiology. Persistent endothelial damage and impaired endothelial regeneration lead to local or systemic endothelial dysfunction and loss of the primary barrier function. Endothelial repair comprises shock, proliferation, acclimation, and endothelial maturation stages. Augmentation of encoding transcripts (FOAS, FOSB, FOSL1 FOSL2, JUNB), stressor protein (activator protein-1, AP-1), and VE-cadherin two hours after endothelial damage are signs of the shock stage. Increased Myc activity, cyclins D1/E2 (cell-cycle genes), and VEGF growth factor initiate proliferative mechanisms. The acclimation process is associated with a robust inflammatory response and remodeling of the basal extracellular matrix. The last phase of endothelial regeneration involves reorganization of the extracellular matrix through the activation and expression of the COL5A1, COL5A2, COL8A1, NID1, and LAMC 3 genes [491]. Additionally, BMP [492,493], hypoxia-inducible factor 1 α (HIF-1α), apolipoprotein (A-I), high-density lipoproteins (HDL), zinc finger transcription factor (ZFP580), syndecan-1, and ixmyelocel-T stimulate endothelial regeneration [494]. Endothelial regeneration can be facilitated by intercellular interplay with surrounding cells (regeneration satellites). Smooth muscle cells, for example, mediate the EC recruitment in a PKCδ-dependent manner by releasing the chemokine (C-X-C motif) ligand 7 (CXCL7) [495]. Angiogenic T-cells and monocytes also contribute to endothelial repair [496,497]. M2 macrophages stimulate endothelial proliferation [498] in lung injury and may attach to endothelial cells in this reaction [498,499,500].

Endothelial turnover, regeneration, and repair are crucial processes for maintaining vascular health throughout life and in the post-injury period. These processes are tightly regulated to ensure proper endothelial function. Quality control mechanisms maintain the functionality of the endothelial layer, degradation processes remove damaged cells, and regeneration and repair mechanisms restore vascular integrity.

These processes must be precisely balanced; dysregulation of any of them leads to the initiation of various pathological conditions, including inflammation, cancer, cardiovascular issues, nervous dysfunctions, etc.

Key endothelial control mechanisms:

(1) Autophagy: removing damaged organelles and proteins [501]; (2) unfold protein response (UPR) is realized when the misfolded proteins are accumulated in the endoplasmic reticulum [502,503]; (3) DNA damage responses (DDR) are crucial for the detection and repair of DNA damage [504,505].

The degradation processes consist of (1) the ubiquitin–proteasome system (UPS) (tags damaged or unnecessary proteins with ubiquitin for degradation) [506,507]; (2) lysosomal degradation (breaks down cellular waste) [508]; and (3) caspase-mediated degradation (activated during apoptosis) [509].

Regeneration mechanisms are composed of: (1) proliferation of existing ECs [510,511]; (2) transdifferentiation (cells change from one type to another and replenish the endothelial layer) [512]; (3) recruiting of circulating cells (circulating cells differentiate into mature ECs to aid in repair) [513].

Repair mechanisms include (1) wound healing (natural response to injury) [514]; (2) shear stress active adaptation (ECs adapt to changes in blood flow) [515]; and (3) EndMT (ECs transform into mesenchymal cells to aid in tissue repair and remodeling) [516].

## Figures and Tables

**Figure 1 cells-13-01276-f001:**
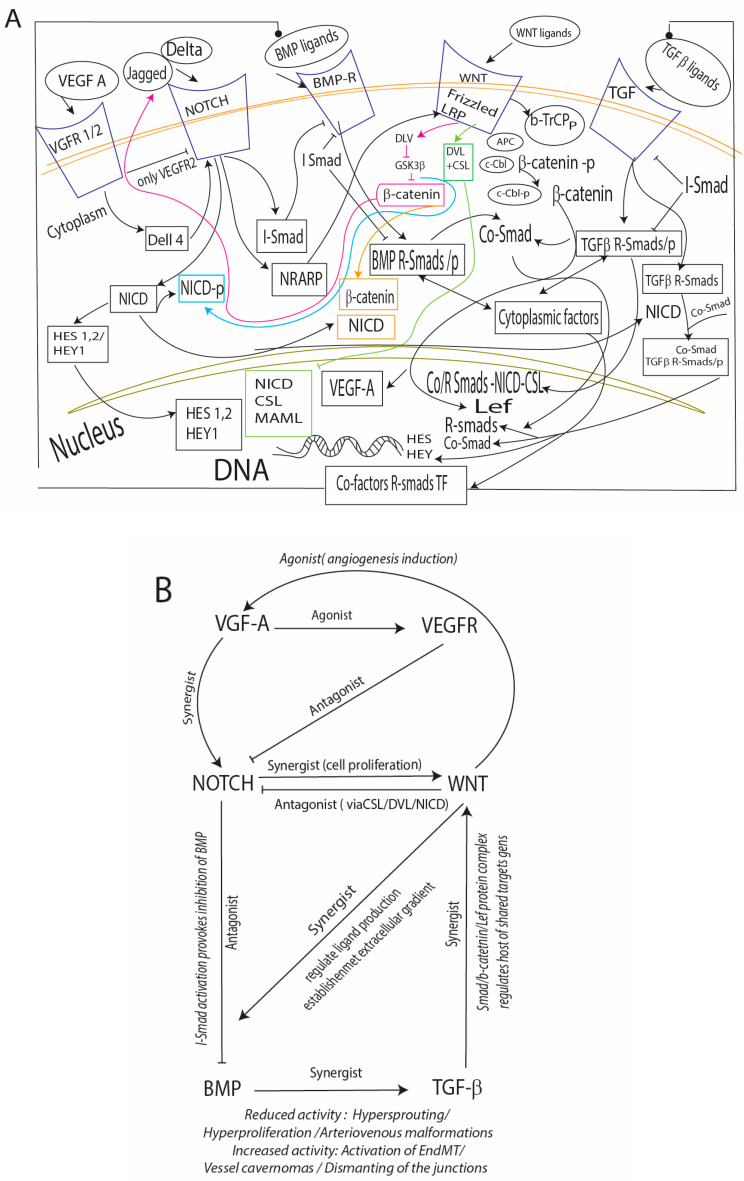
(**A**). Interaction and cross-talk between different pathways in endothelial development and function. Molecular signaling cross-talk: WNT/VEGF-A: in ECs, WNT-pathway is constitutively suppressed in ECS. WNT-ligand activates FRZ receptors through the inactivation of the destruction complex (APC-c-Cbl β-TrCP- β-catenin) and translocation of β-catenin to the nucleus with further activation of VEGF-A; VEGF/NOTCH demonstrates a synergistic effect. VGF, through the VEGFR2, increases expression of DLL4, leading to NOTCH activation. NOTCH receptors demonstrate synergism (VEGF 1, VECGF3) and antagonism (VEGF2) in VEGF expression; NOTCH/BMP activates expression of the inhibitory I-SMAD protein, leading to inhibition of BMP 2/6; NOTCH/TGF-β cross-talk is characterized by downstream of ALK receptors and depends on R-SMAD and Co-SMAD activation with the formation of SMAD/NICD complex of transcription (similar to β-catenin/NCID complex). As a result, the FGF-β-induced expression of NOTCH target genes (HEY/HES) leads to migrations of ECs; BMP/WNT/TGF-β demonstrate synergetic regulation and determine ligand production of each other; WNT/TGF-β cross-talk leads to synergetic regulation of the set of shared target genes in the nucleus via the Smad/Lef/β-catenin. NOTCH/WNT cross-talk is complex and involves several regulation levels: formation of β-catenin/NICD transcriptional complex (orange lines), the interplay between NOTCH receptors and β-catenin at the membrane (pink lines), NICD phosphorylation by GSK3b (blue lines), and inhibitory interaction between CSL and Disheveled (DVL) and inhibition of effector gene expression (green). (**B**) Schematic representation of the pathway signaling cross-talk in ECs: APC—adenomatous polyposis coli; BMP—bone morphogenetic proteins pathway; BMP ligands: BMP 2,4,5,6,7; BMPR—bone morphogenetic proteins receptor; b-TrCP (Fbxw1 or hsSlimb)—β-transducin repeat-containing protein; b-TrCP-p—phosphorylated b-TrCP; c-Cbl—casitas B lineage lymphoma protein with Ee ligase activity; c-Cbl-p—phosphorylated c-Cbl; Co-SMADS: SMAD 4; CSL-(CBF1, suppressor of hairless, Lag1)—transcription factor activating the genes downstream of the NOTCH pathway; HES 1,2—hairy and enhancer of split 1,2; HEY 1—hairy/enhancer of split related with YRPW motif protein 1; I-SMADS: SMADS 6, 7; NICD—NOTCH intracellular domain; R-SMADS: SMADS 1,2,3,5,8; TGF-β—transforming growth factor β; TGF-b ligands: TGF-b 1. 2,3; VEGF-A—vascular endothelial growth factor A; VEGFR—vascular endothelial growth factor receptors; WNT—wingless-related integration site.

**Figure 2 cells-13-01276-f002:**
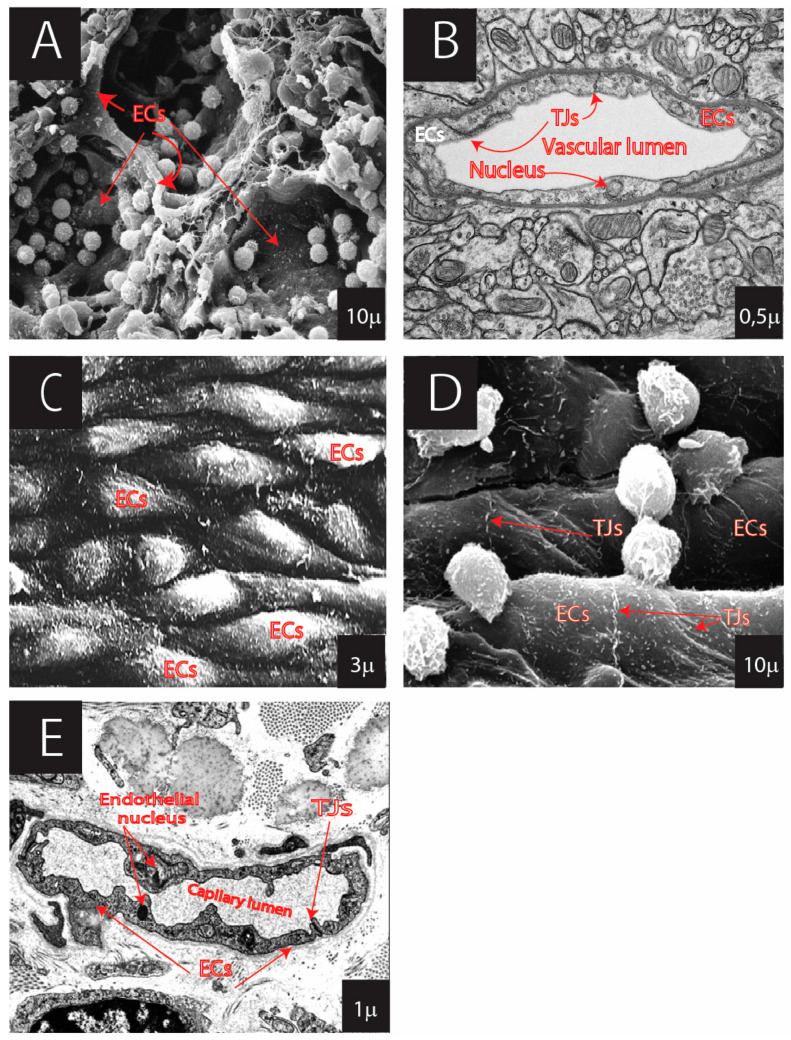
Continuous ECs showing diversity in various organs: (**A**) medullary sinus of lymph node (SEM: surface scanning electron microscopy); (**B**) brain with tight junctions (TJ) of the blood–brain barrier (TEM: transmission electron microscopy); (**C**) ECs of arteries (SEM); (**D**) ECs of veins (SEM); (**E**) EC of capillary of dermis/skin (TEM). Kindly provided by emeritus professor P. Groscurth through https://e-learn.anatomy.uzh.ch/Anatomie/Anatomie.html.

**Figure 3 cells-13-01276-f003:**
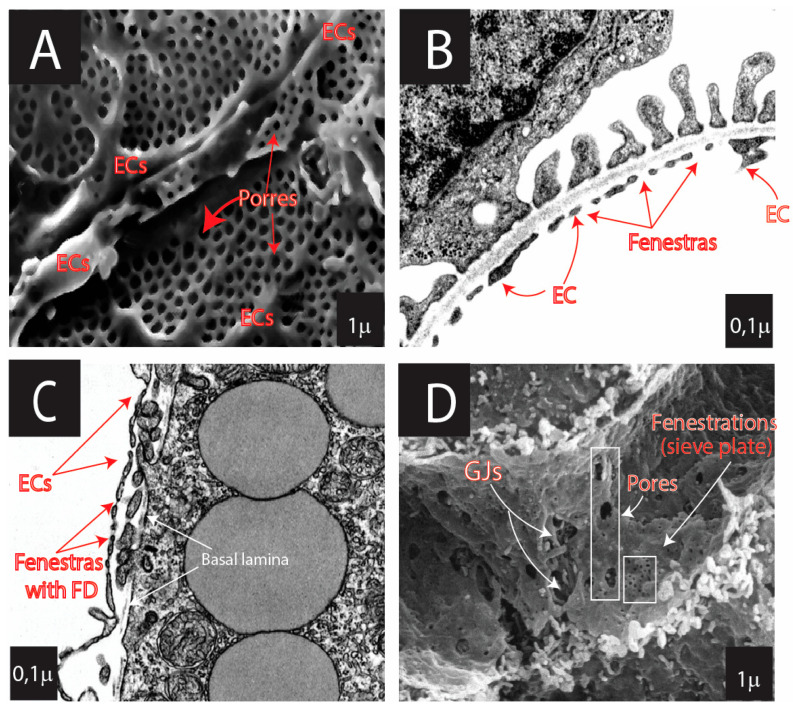
Fenestrated ECs showing diversity in the organs: (**A**) ECs of kidney glomerulus (SEM); (**B**) ECs of kidney glomerulus (TEM); (**C**) adrenal gland capillary with fenestrated ECs and fenestral diaphragms (FD) (TEM); (**D**) ECs of the sinusoid of liver (SEM). (Kindly provided by emeritus professor P. Groscurth through https://elearn.anatomy.uzh.ch/Anatomie/Anatomie.html).

**Figure 4 cells-13-01276-f004:**
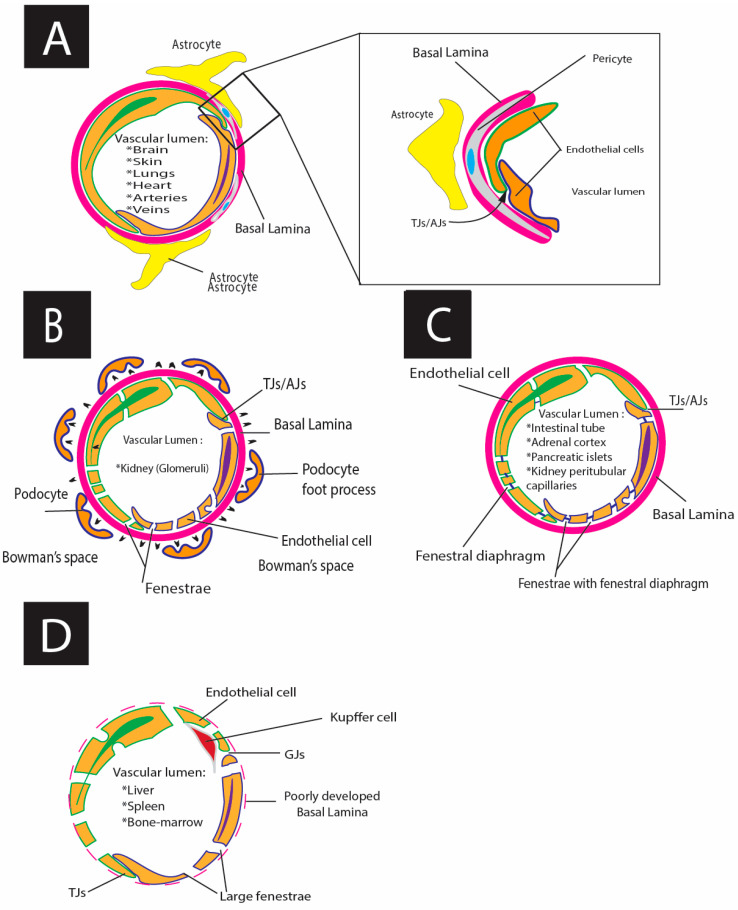
Schematic representations of different types of endothelium: (**A**) Schematic drawing of continuous endothelium; (**B**) schematic drawing of fenestrated endothelium (glomerular fenestrated endothelium (true fenestrated), with fenestrae ~60–100 nm in diameter), TJ (tight junctions), and AJ (adherens junctions); (**C**) schematic drawing of pseudo fenestrated endothelium with fenestral diaphragm (FD (fenestrae ~60–70 nm in diameter); and (**D**) schematic drawing of discontinuous sinusoid endothelium with GJs (gap junctions), TJs, and large pores ~100–200nm.

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
