# Peer review of "Dynamics of Endothelial Cell Diversity and Plasticity in Health and Disease"

_cells, 2024, doi:10.3390/cells13151276_

Round 1

Reviewer 1 Report

Comments and Suggestions for Authors

The endothelium is the biggest organ in the human body, therefore it plays a very important role in the maintenance of the cardiovascular system fitness.

The presented manuscript summarizes very deeply the current knowledge about endothelial diversity and plasticity, more than 480 papers are cited. However, the authors mainly focus on the details concerning tissue structure and indicate specific protein markers for endothelium in particular organs.

The manuscript is well-written and illustrated with four figures. Some details are summarised in tables, which make it easier to understand.

Minor comments:

1. A few sections are hard to read and understand as they contain many unexpanded abbreviations. Finally, they are described in further paragraphs but in general, it should be done with the first use in the text.

2. Figure 1: there are a lot of details and finally the figure is illegible.

3. Line 32:  13 should be in superscript.

4. In paragraph 4 some pieces of information, that appear in paragraph 2.1, are repeated (concerning NOTCH etc.).

5. Figure 2E: the inscriptions are barely visible.

6. Line 465: in which unit thickness is given?

7. In paragraph 6, pieces of information concerning the characteristic structure of a particular organ given in paragraph 5, are repeated. In my opinion, paragraph 6 summarizes more general facts, so it should be found first than paragraph 5.

8. In the case of endothelial turnover, there is no information about quality control mechanisms, degradation processes etc.

The manuscript gives a very good background for everyone who wants to expand their knowledge concerning the intercellular junctions in endothelium and the existence of differences between different organs directly linked with the function. Moreover, the authors indicate specific protein markers which help to characterize the tissue origin. In my opinion, this manuscript will be suitable for publication in Cells after considering the above-mentioned comments. 

Author Response

Dear Colleague, Dear Reviewer 1,

First of all, we would like to thank you for your time and interest in our work. We appreciate your detailed analysis, which has helped us improve our review. Please find our responses to your comments below. All changes here are highlighted in red.

Comment 1: 1. A few sections are hard to read and understand as they contain many unexpanded abbreviations. Finally, they are described in further paragraphs but in general, it should be done with the first use in the text.

Response: Thank you for this valuable suggestion. We agree with your comment and have made the following changes:

   1) Many abbreviations have been removed from the text.

   2) The retained abbreviations are now accompanied by their full terms at their first mention, e.g., intraembryonic hemogenic endothelial cells (IHECs), fibroblast growth factors (FGFs), etc.

    3) We have added a list of abbreviations at the end of our review (page 33 in the manuscript).

ABBREVIATIONS

ANG1 and ANG 2-angiopoietins 1 and 2

ACE1-Angiotensin 1-converting enzyme

BMP- bone morphogenetic proteins pathway

DNA-deoxyribonucleic acid

ECM-extracellular matrix

ECs-Endothelial cells

EMT - epithelial-mesenchymal transition

EndMT-endothelial-to-mesenchymal transition

ER71 (ETV2) - ETS variant transcription factor 2

ESAM-Endothelial cell-selective adhesion molecule

ETS family –E twenty-six family transcription factors

FGF -fibroblast growth factor

FLI1-friend leukemia integration-1 transcription factor

HES1, 2-hairy and enhancer of split 1, 2

HEY1-hairy/enhancer of split related with YRPW motif protein 1

HGF- hepatocyte growth factor

HIF-1a- hypoxia-inducible factor 1a

Hh -Hedgehog

ICAM-1 -intercellular adhesion molecule-1 (or CD54)

IHECs-intraembryonic hemogenic endothelial cells

LMO2-Lim domain only 2

LSECs-liver sinusoid ECs

LYVE 1-lymphatic vessel endothelial hyaluronan receptor

MAML1,2,3- mastermind like protein 1,2,3

MAP4K4 -mitogen-activated protein- 4- kinase 4

NRP 1,2 -neuropilin1, 2

PI3K/Akt phosphoinositide 3-kinase / protein kinase B

PDGF - Platelet-derived Growth factor

PDGFR-b-Platelet-derived Growth factor receptor-b

RNA-ribonucleic acid

RUNX1-runt-related transcription factor 1

S1PR1sphingosine 1-phosphate (S1P)- sphingosine 1-phosphate receptor

SCL/TAL1 or TAL1 or SCL- stem cell leukemia/or T-cell acute lymphocytic leukemia-1

scRNA-seq analysis – single-cell RNA sequencing analysis

SFRP-secret frizzled-related protein

SMADs homologies to SMA ("small" worm phenotype) and MAD family ("Mothers Against Decapentaplegic") genes

SSECs-spleen sinusoid ECs

TAK-1 -transforming growth factor-beta activated kinase 1

TCF7L2-transcription factor 7 like 2 (TCF4)

TGFb ,a -transforming growth factor b

TIE-tyrosine kinase with immunoglobulin-like and EGF-like domains

TLR2 -toll-like receptor-2

TNF-a-tumor necrosis factor-a

TWIST1 -twist family of basic helix-loop-helix protein 38 (bHLHa38) transcription factor 1

VCAM1-vascular cell adhesion protein1

VE-cadherin -vascular endothelial cadherin (cadherin-5 or CD144)

VEGF -vascular endothelial growth factor

VEGFR-1 (Flt 1)-vascular endothelial growth factor receptor 1

VEGFR2 (Flk1/KDR) -vascular endothelial growth factor receptor 2

vWf -von Willebrand factor

WNT- Wingless-related integration site

Comment 2. Figure 1: there are a lot of details and finally the figure is illegible.

Response: We agree with the comment. The original figure has been divided into two separate figures: Figure 1A (page 16 in the manuscript, line 249) and Figure 1B ( page 17 in the manuscript, line 278). These two distinct parts better reflect the complex regulation of endothelial biology. Additionally, we have improved the legibility of the text by increasing the font size. However, if this approach is not satisfactory, we can delete Figure 1A and retain Figure 1B as a schematic summary of the regulation of endothelial biology. Please find new pictures 1A and 1B below:

Figure 1A. Interaction and cross-talk between different pathways in endothelial development and function: (a) Molecular signaling cross-talk: WNT/VEGF-A: in ECs, WNT-pathway is constitutively suppressed in ECS. WNT-ligand activates FRZ Receptors through the inactivation of the destruction complex (APC-c-Cbl b-TrCP- b-catenin) and translocation of b-catenin to the nucleus with further activation of VEGF-A; VEGF/NOTCH demonstrate synergistic effect: VGF through the VEGFR2 increases expression of DLL4 leading to NOTCH activation. NOTCH receptors demonstrate synergism (VEGF 1, VECGF3) and antagonism (VEGF2) in VEGF expression; NOTCH/BMP activated expression of the inhibitory I-SMAD protein leading to inhibition of BMP 2/6; [NOTCH/TGF-b cross-talk characterized by downstream of ALK receptors and depends on R-SMAD and Co-SMAD activation with the formation of SMAD/ NICD complex of transcription (similar to b-catenin /NCID complex). As a result, the FGF-b induced expression of NOTCH target genes (HEY/HES) leads to migrations of ECs; BMP/WNT/ TGF-b demonstrate synergetic regulation and determines ligand production of each other; WNT/ TGF-b cross-talk leads to the synergetic regulation of the set of shared target genes in the nucleus via the Smad/Lef/b-catenin NOTCH/WNT cross-talk is complex and involves several regulation levels: formation of b-catenin/NICD transcriptional complex (orange lines), the interplay between NOTCH receptors and b-catenin at the membrane (pink lines), NICD phosphorylation by GSK3b (blue lines) and inhibitory interaction between CSL and Dishevelled (DVL) and inhibition of effector gene expression (green).

Figure 1B. Schematic representation of the pathway signaling cross-talks in ECs: APC-adenomatous polyposis coli; BMP- bone morphogenetic proteins pathway; BMP ligands: BMP 2,4,5,6,7; BMPR- bone morphogenetic proteins receptor; b-TrCP ( Fbxw1 or hsSlimb) -b-transducin repeat-containing protein; b-TrCP-p -phosphorylated b-TrCP; c-Cbl- casitas B lineage lymphoma protein with Ee ligase activity; c-Cbl-p-phosphorylated c-Cbl; Co-SMADS: SMAD 4; CSL-(CBF1, suppressor of hairless, Lag1) - transcription factor activating the genes downstream of the NOTCH pathway; HES 1,2- hairy and enhancer of split 1,2; HEY 1- hairy/enhancer of split related with YRPW motif protein 1; I-SMADS: SMADS 6, 7; NICD - NOTCH intracellular domain; R-SMADS: SMADS 1,2,3,5,8; TGF- b transforming growth factor b; TGF-b ligands: TGF-b 1. 2,3; VEGF-A - vascular endothelial growth factor A; VEGFR- vascular endothelial growth factor receptors; WNT- Wingless-related integration site

Comment 3. Line 32:  13 should be in superscript.

Response: We agree with the comment and have made the necessary changes, page 1, line 33 in the new manuscript

Comment 4. In paragraph 4 some pieces of information, that appear in paragraph 2.1, are repeated (concerning NOTCH etc.).

Response: We agree with the comment. We have rewritten paragraph 4.1 ( page 19, line 377 in the version of the manuscriot) to focus on the target regulation of endothelial plasticity and heterogeneity. Specific information regarding the NOTCH, WNT, and VEGF pathways has been added in the context of the regulatory mechanisms of endothelial plasticity and heterogeneity. Additionally, new pathways related to the regulation of plasticity, such as LMO2 and TWIST2, have been included. Please see the changes highlighted in red below.

4.1 Target regulation of endothelial plasticity and heterogeneity

Not much is known about the factors that may determine endothelial plasticity.

The pathways thoroughly described before play a crucial role in this process:  NOTCH ( in general described before), WNT( mentioned above), BMP ( described before), TGF, (VEGF) [215], Lim domain only 2 (LMO2) [216], ETS variant transcription factor 2 (ETV2) [217], twist family BHLH transcription factor 1 (TWIST-1) [218], hypoxia-inducible factor 1 (HIF1)[219].  This section specifically focuses on the characteristics associated with endothelial plasticity and heterogeneity.

NOTCH

NOTCH is one of the principal regulators of endothelial heterogeneity, determination, and endothelial apoptosis. NOTCH as was mentioned before inhibits the proliferation of the sprouting ECs through the direct interaction between the receptors and DLL4 [220]. Endothelial plasticity and heterogeneity are closely related to NOTCH 1 and NOTCH 4 receptors. They differently influence endothelial apoptosis and contribute to the complex regulation of vascular remodeling. Thus, NOTCH1 and NOTCH4 determine apoptosis and normal morphology and are related to cellular context and external stimuli. Activation of NOTCH1 signaling leads to an augmentation of cell survival under stress conditions via upregulation of anti-apoptotic proteins Bcl-2 and Bcl-xl [221]. Additionally, NOTCH 1 signaling in cell survival may enhance the expression of survival pathways such as PI3K/Akt [222]. Notch4 signaling is an antagonist of NOTCH1 under certain conditions. Overexpression of NOTCH4 leads to enhanced endothelial apoptosis through the induction of pro-apoptotic factors like p53 and Bax [223]. NOTCH4 leads to the dynamic restriction of vascular networks during development, contributing to the pruning of aberrant vessels[224]. NOTCH signaling participates in the transformation of the venous ECs into artery ECS via the NR2F2 (Nuclear Receptor Subfamily 2 Group F, Member 2 ) receptor [225].

WNT

WNT is one of the crucial regulators of endothelial plasticity and heterogeneity. As mentioned before, WNT signaling acts 1) all over the body through the canonical WNT/catenin  signaling [226–228] and non-canonical pathway WNT/Ca2+ [229,230] and 2) locally in context-related responses [231,232]. WNT determines endothelial morphology in a case of disturbed blood flow, controls endothelial integrity and regulates the distribution of tight junctions[233], and controls endothelial maturation[234]. All the aforementioned mechanisms may be considered as the base of endothelial heterogeneity.

BMP

BMP signaling selectively influences ECs in a context-dependent manner without affecting endothelial survival through SMADS effectors and a non-canonical pathway via sp38, MAPK, and ERK [235]. The transcriptional activation of BMP signaling occurs under distinct epigenetic influences. For example, the turbulent shear stress induces phosphorylation of SMAD1/5/8, which provokes an activation and proliferation of ECs. Laminar shear stress leads to endothelial hibernation through the BMP9/Alki1/ENG-SMAD cascade [236,237]. Interestingly, SMADs may also interrelate with the p300/CREB binding protein (CBP) signal transduction and readily create a complex with the NOTCH intracellular domain, activating NOTCH target genes. Posttranscriptionally, BMP targets are related to reversible phosphorylation process [237,238], ubiquitination, and sumoylation [237,239] affecting cell survival [240,241] .

TGFb

The transforming growth factor-(TGF signaling pathway plays a vital role in the regulation of EC heterogeneity of the vascular systems. Different aspects of endothelial biology are influenced by TGFThis pathway signals through the type I and II serine/threonine kinase receptors: ALK1 (leading to SAMAD1/5/ activation and promotion of the EC proliferation and migration) and ALK5 (phosphorylation of SMAD2/3 leading to inhibition of  EC proliferation and migration) [242,243]. The role of TGF is that of a potential inducer of EndMT through the SMAD-dependent and SMAD-independent pathways[244,245].

VEGF

 VEGF can be considered a driver of endothelial plasticity due to the heterogeneity of ECs. Deletion or mutation of the VEGFR1 had different effects in different organs: in the retina, it increased vascular density, and in the liver, there is no change [235]; deletion of the VEGFR2 causes regression of the vascularization [235], and VEGFR3, similar to VEGFR1, initiate hypersprouting of vessels in the retina [246]. It is interesting that different VEGF isoforms determine the phenotypic specialization of endothelial cells in different organs [247]. This explains the phenomena when ECs of various organs respond differently to VEGF changes in the vascular bed[248].

LMO2

LMO2 is a crucial element in the axis of LMO2 -Prdm16 (PR domain containing 16) involved in the development of hematopoietic stem cells through the endothelial-to-hematopoietic transition as a form of endothelial plasticity   [249]. Moreover, in the zebrafish, LMO 2 is a main fate plasticity messenger that determine the fate- decision between endothelial and pronephron tubule cells  [250].

ETS (ETV2)

ETS2 plays a pivotal role in endothelial plasticity in the context of specification and early differentiation from mesodermal progenitors[251]. This transcription factor has several lines of action and participates in the migration of progenitor ECs [252] and regulates endothelial reprogramming[232,253].

ETV2 is able to directly  reprogramme cells ( shown on fibroblasts) into endothelial fate line[254] and upregulates cell migration through increase of  gene chromatin accessibility regulating cell migration [255].

TWIST-1

 The role of this transcriptor factor is strongly associated with epithelial-to-mesenchymal transition (EMT) and is well studied mainly in tumorogenesis [256,257]. Endothelial cells actively participate in tumor neovascularisation and exhibit high levels of plasticity and heterogeneity during this process. TWIST1 stimulates EMT by representing E-cadherin expression and upregulating mesenchymal markers such as vimentin[256,257].

HIF1

As a universal protective molecule aganst  low oxygen concentrations, HIF1 is involved in the induction of  VEGF [258], a nd the modification of the gene expression that promotes angiogenesis  and vascular remodeling[259]. Additionally, HIF1 is potentially linked with  EndMT and may influence cancer  and other pathological conditions[259].

To summarize, the concept of endothelial plasticity is not only an ability of ECs to adapt their own initial genetically determined programs to the local epigenetic landscape but an ability to induce plasticity of surrounding cells belonging to different cell types. ECs are able to build an endothelial-friendly local environment [260–263], e.g., co-culturing ECs with tanycytes induces cytoskeleton remodeling in the latter through the NO-mediated signaling pathway with cGMP and involvement of prostaglandins-thromboxanes [264]. Herein, ECs indirectly control the glial-neuronal interactions in the brain. To summarize, the concept of endothelial plasticity is not only an ability of ECs to adapt their own initial genetically determined programs to the local epigenetic landscape but an ability to induce plasticity of surrounding cells belonging to different cell types. ECs are able to build an endothelial-friendly local environment [247–250], e.g., co-culturing ECs with tanycytes induces cytoskeleton remodeling in the latter through the NO-mediated signaling pathway with cGMP and involvement of prostaglandins-thromboxanes [251]. Herein, ECs indirectly control the glial-neuronal interactions in the brain.

Comment 5. Figure 2E: the inscriptions are barely visible.

Response: We agree with the comment. Figure 2 has been completely revised and its legibility has been improved. Please find the updated Figure 2 below (page 22 , line 472 in the new version of the manuscript):

Figure 2.   Continuous ECs showing diversity in various organs: A) medullary sinus of lymph node (SEM: surface scanning electron microscopy); B) brain with tight junctions (TJ) of the blood-brain barrier (TEM: transmission electron microscopy); C) ECs of arteries (SEM); D) ECs of veins (SEM); E) EC of capillary of dermis/skin (TEM). Kindly provided by emeritus professor P. Groscurth through https://e-learn.anatomy.uzh.ch/Anatomie/Anatomie.html

Comment 6. Line 465: in which unit thickness is given?

 Answer: in micrometers , mm, please look line 720 ( the previous number was 465), page 29 in the new version of manuscript.

Comment 7. In paragraph 6, pieces of information concerning the characteristic structure of a particular organ given in paragraph 5, are repeated. In my opinion, paragraph 6 summarizes more general facts, so it should be found first than paragraph 5.

Response: We agree with your suggestion. Thank you for your proposition. We have changed the order of the sections: Section 6 has been moved to replace Section 5. Please find the changes implemented below the titles in red(page 23 ,paragraph 5 in the manuscript, lines 494-678// page 28 for paragraph 6, lines 678-817)

  1. Endothelial diversity along the vascular bed

5.1 Diversity of ECs in the microcirculation

ECs at the microcirculation level (arterioles, capillaries, post-capillary venules with a lumen diameter of less than 100 mm) demonstrate more substantial heterogeneity than the macrovessels  [265–267]. Arterial, venous, and lymphatic micro/macrovessel ECs across the same tissue have similar morphology and demonstrate analogous transcriptomes. Despite standard endothelial features, the transcriptomes in different tissues are also entirely different [268].  Local epigenetic factors that influence microvascular diversity are shear stress, intravascular pressure, and metabolic clues [269]. Genetic influence is essential for the biochemical organization of the endothelial basal lamina, a thin sheet of extracellular matrix (ECM) located at the epithelial cells’ basal surface. It can be visualized by light microscopy and consists of a lamina lucida or lamina rara (composed of laminin, integrins, entactins, dystroglycan), a lamina densa (collagen IV type, perlecan), and a lamina fibroreticularis. Laminae lucida and lamina densa are named in the literature as the basal lamina and can be visualized only electron microscopically [270–272]. The specific microvascular gene expression patterns are primarily implicated in producing particular basal lamina proteins, including laminin, collagen (4a1 and 4a2), and ECM-interacting proteins (cD36, a1 integrin, a1 integrin, b4 integrin). The microcirculation-specific genes regulate the individual biochemical characteristics of the endothelial cell membrane. Microcirculatory ECs have a surface covered with a negatively charged glycocalyx (composed of glycoproteins and glycolipids). Glycocalyx thickness varies between different parts of the microcirculatory tree [273,274]. The ECs of the fenestrated glomerulus have a thick and specialized glycocalyx (e.g. playing a role in the filtration barrier) whereas the ECs of the fenestrated sinusoidal capillaries of the liver have a thin glycocalyx layer, allowing quick exchange between the blood and space of Disse [275].

At the microcirculatory level, endothelial morphology is strongly correlated with its functions. ECs are divided into several phenotypes: 1) continuous (have continuous basal lamina and have no fenestrations) (Figure 2, Figure 4); 2) fenestrated (have fenestrations) with several subtypes (Figure 3, Figure 4): a) pseudo-fenestrated (fenestrations covered by fenestral diaphragm ), b) glomerular fenestrated (without fenestral diaphragm); c) discontinuous fenestrated cells (lack of basal lamina) ( Figure 3)[26,276–309].

Figure3. Fenestrated ECs showing diversity in the organs:  A) ECs of kidney glomerulus (SEM); B) ECs of kidney glomerulus (TEM); C) adrenal gland capillary with fenestrated ECs and fenestral diaphragms (FD) (TEM), D) ECs of the sinusoid of liver (SEM). (Kindly provided by emeritus professor P. Groscurth through https://elearn.anatomy.uzh.ch/Anatomie/Anatomie.html.)

Continuous ECs are present in tissues with a low but well-controlled exchange. This endothelial cell type’s primary characterization is the continuous basal lamina and lack of fenestra (Figure 2, Table 2) [310]. In contrast, fenestrated and sinusoidal endothelium is mainly located in the organs with a high uni- or bidirectional substrate diffusion.

Pseudo-fenestrated ECs are located in endocrine tissues, gastrointestinal mucosa, and renal peritubular capillaries. They have a continuous basal lamina and pores (60-70 nm in diameter) with a thin diaphragm (Figure 3, Table 2).

   Discontinuous fenestrated ECs (or sinusoidal ECs) are characterized by a lack of a continuous basal lamina, multiple fenestrations with 50-100 nm in diameter aggregated into a group of 10-100 fenestrae (liver “sieve plates”), and big pores (100-200 nm in diameter) without diaphragm. These types of ECs are typical for the liver, spleen, and BM (Figure 4, Table 2).

Figure 4. Schematic representation of different types of endothelium:(a) Schematic drawing of continuous endothelium, (b) Schematic drawing of fenestrated endothelium (glomerular fenestrated endothelium (true fenestrated), with fenestrae ~60-100nm in diameter), TJ (tight junctions), AJ (adherens junctions), (c) Schematic drawing of pseudo fenestrated endothelium with fenestral diaphragm (FD (fenestrae ~60-70 nm in diameter), (d) Schematic drawing of discontinuous sinusoid endothelium with GJs (gap junctions) and TJs, and large pores ~100-200nm.

Glomerular fenestrated ECs take the intermediate position between common pseudo-fenestrated and discontinuous fenestrated ECs: they have a basal lamina, pores with a diameter 60-80, and no diaphragms (Table 2) [303,311,312]. The local microcirculatory functions and morphology contribute to permeability regulation, i.e. an exchange between the vascular lumen and the extravascular tissue [269,313,314]. The above-mentioned phenotypic differences of ECs directly affect microcirculatory endothelial permeability [315].

High endothelial venules (HEVs) are a special part of the microcirculatory bed[316]. These vessels belong to post-capillary venules, which mostly accompany lymphoid tissues such as lymph nodes, Peyer’s patches, and tonsils. The specificity of these cells is strongly related to the immune system functioning and serves to facilitate the entry of lymphocytes from the bloodstream into the lymphoid tissues( place of immune response initiation and regulation). The ECs of  HEVs have a flat and cuboidal shape, unlike the flat and elongated morphology of typical ECs. Notably, the surface of these types of ECs displays a specific glycosylation pattern composed of  Pereferial node addressins (PNAd) and express high levels of sialomucins which contribute to lymphocyte homing through binding L-selectin on lymphocytes[317]. The listed structural and functional characteristics facilitate the mediation of lymphocyte trafficking into lymphoid tissues, supporting effective immune surveillance and responses[318].

Critical structural elements of the endothelia, regulating the cellular diffusion process (especially across the continuous endothelium) and layer stability, are cell-to-cell junctions [319]. Junctional endothelial diversity is another marker of EC heterogeneity, characterized by variations in the expression of junctional proteins[320]. The junctions connect to cytoskeletal and several signaling proteins that contribute to the maintenance of shape and polarity [321,322]. There are three types of cellular junctions: tight junctions (TJs), adherence junctions (AJs), and gap junctions (GJs) [278].

TJs are associated with barrier tissues, where the fast interchange between the blood and tissues does not exist or is strictly regulated. These types of junctions are typical for the endothelium in the brain, the lungs, endocardium, nerve capillaries, fat, and muscle tissue. They display the role of a selective wall allowing for the select paracellular passage of ions between cells [323,324]. The claudin family (26 proteins in humans), occludin, and tricellulin are responsible for the TJs barrier specificity [325,326]. In these tissues,  TJs are yet not only responsible for the barrier function (blood-brain-barrier, air-blood barrier, blood-nerve barrier, but also for the physical stabilization of macrostructures with regular physical stress, e.g. muscle and endocardial ECs adapt quickly to shape-changing during muscle contraction [327–331].

AJs are cell-cell adhesion complexes interacting with the F-actin cytoskeleton and contribute to tissue homeostasis, embryogenesis, stabilization, initiation of cell-cell adhesion, and control of intracellular signaling. These multiprotein complexes are composed of cadherins and nectins [332]. TJs and AJs are zipper-like elements localized at the lateral membrane of the ECs [333]. The function and stability of EC AJs are regulated by various types of interactions with actin cytoskeleton participating in modulation of endothelial permeabilityas a reaction on different types of stimuli [334].

GJs are members of the connexin protein family, which form intercellular channels and provide direct communication patterns between endothelial and surrounding cells: electrical coupling and flow of metabolites in exchange [335]. GJs proteins are predominantly tightly associated with the plasma membrane and, additionally to being passively transported, may also play a role in the rough endoplasmic reticulum, mitochondria, and Golgi apparatus  [336]. GJs participate in transmitting vasodilatory signals from the capillary network to arterioles as well as conduct signals from the endothelial to muscle cell layer in some vascular beds [337,338]. They contribute to thechronic remodeling of vessels by induction of cellular stiffness, actin rearrangement, and activation of pro-inflammatory genes that result in disease development [339,340]. GJs are mainly located in organs with a rapid molecular exchange between the blood and the surrounding compartment (endocrine glands, lymphatic capillaries, liver sinusoids, spleen, and BM). GJs facilitate fast information exchange that helps the whole endothelium to respond to a focal signal in a coupled fashion, like a syncitium [63,341–343].

5.2 Diversity of ECs in the large blood vessels

Expression of genes specific for the macrocirculation bed contributes to modulation and biosynthesis of corresponding ECM: fibronectin, collagen (5a1 and 5a), and osteonectin. [344]. The main epigenetic microcirculatory factors are arterial pressure, hypercholesterolemia, and inflammation [345–347].

Morpho-functionally, arterial ECs (AECs) and venous ECs (VenECs) specialize in the non-stop conduction of oxygen-rich/poor blood towards or away from the heart. Macrocirculatory ECs participate in the systemic control of blood pressure, regulation of blood flow (shear stress) and vasomotor tone (circumferential wall stress). Therefore, they are also involved in the maintenance of the total peripheral vascular resistance (elastance and compliance) and vascular capacitance,  activation and migration of blood cells/ immune cells, as well as in vascular disease development [269,313,314]. AECs have elongated and narrow shapes. They contribute to the tunica intima, which is reinforced by smooth muscle cells. In comparison with VenECs, AECs do not participate in the formation of valves. Additionally, AECs provide fewer conduits for transmigration of immune cells [348,349] and play a lesser role in terms of inflammation, when comparedwith the venous endothelium [350]. Arterial and venous ECs belong to the continuous endothelial cell type. AECs and VenECs contain TJs and AJs, which provide cell-cell interactions and secure vessel stability (providing laminar blood flow and reducing vascular shear stress).

5.3 Lymphatic macro/microcirculation

The lymphatic system belongs to the drainage system of the body, closely related to the blood circulation [351]. It furnishes the peripheral immune system [352] and nervous system with drainage of the extracellular liquid and corresponding clearance of antigens contributing to the maintenance of brain homeostasis [353,354]. The lymphatic system’s essential functions include the removal of interstitial fluid and the maintenance of local homeostasis, local tissue immunological supervision, and host defense which entails cell trafficking, transcytotic delivery in the guidance and support of immunocompetent cells. It also is involved in the absorption of dietary lipids and participates in the vessel and organo-vascular morphogenesis [355]. The morpho-physiological organization of the lymphatic bed determines its functionality.

The lymphatic capillaries consist of a single layer of lymphatic ECs (LECs) characterized by different cellular integrity levels and the absence of a basal lamina in contrast to most blood vessels [356]. The “initial blind-ended” capillaries eliminate intercellular overflow and regulate the macromolecular balance in the interstitial space. LECs in the initial capillaries interconnect directly with the interstitial matrix, forming the specific discontinuous oak-leaf shape, button-like junctions (buttons) [357]. The larger collecting capillaries drain into the thoracic or right lymphatic duct and, finally, to the brachiocephalic veins. The level of cell-cell integrity between “collecting” LECs is higher as compared to the “initial” LECs, which closely cooperate with smooth muscle cells or pericytes, build continuous zipper-like junctions (zippers), and additionally acquire the intraluminal valves. This architecture provides permanent unilateral lymphatic flow [358]. Ontogenetically, the zippers are older than the buttons [359]. Both junction types demonstrate a high capacity for plasticity under various physiological and/or pathophysiological demands; mature junctions are able to reorganize junctional proteins and the cytoskeleton [289]. Inflammation and infection stimulate the transformation of the existing button-like junctions into zipper junctions [308].  Organ-specific functions may be determined by the source of lymphatic origin and cellular microenvironment  [355]. In early development, the lymphatic vessels arise mainly from large veins. In the postnatal period, lymphatic development occurs by reorganizing lymphatic capillaries or by transdifferentiation of venous, mesenteric, and hemogenic endothelium [360]. Notably, human and mouse myeloid lineage cells can successfully transdifferentiate  to LECs through the activation of toll-like receptor-4 (TLR4) [361].  PROX1, SOX18, NOTCH, Wingless-related integration site (WNT), COUP transcription factor 2, AM-CRL-RAMP2, angiopoietin-TIE, VEGF, VEGFR3, and sphingosine 1-phosphate (S1P)- sphingosine 1-phosphate receptor (S1PR1) participate in the control of lymphatic fate [362–366]. The VEGFR3, VEGFC, PROX1, NOTCH, LYVE l are responsible for the lymphatic migration and sprouting; the specification of LECs is derived by WNT, BMP, JAGGED1/NOTCH1 signaling pathways [367]. The angiopoietin-TIE is necessary for cardiovascular and lymphatic development including remodeling [368,369]. S1P and S1PR1 also participate in LEC remodeling, development, lymphatic valve formation, endothelial barrier function, dilatation of vessels, and inflammation [370].  The dysfunction of the lymphatic system may result in the initiation and progression of a wide range of diseases. Decreased lymphatic drainage and cerebrospinal fluid flow aggravate glioblastoma progression and may influence the onset of pre-senility in the elderly [354,371]. Metabolic disorders such as diabetes mellitus, obesity, and metabolic syndrome are closely related to the dysfunction of the lymphatic system. They are characterized by chronic inflammation, low inflammatory answer, development of secondary lymphedema, and lipedema [372,373].

  1. Selected tissue-specific endothelial phenotypes

The brain, heart, lungs, liver, kidneys, gut, and endocrine organs have distinct endothelial subpopulations displaying specific characteristics.

Brain endothelial cells (BECs), along with pericytes and astrocytes, form the blood-brain barrier and enforce the protective barrier function [374]. The relationships between pericytes and ECs (including BECs) are dynamic and vital for the integrity and maintenance of vessel walls [375]. Functionally, pericytes play a vital role in modulating blood flow, supporting angiogenesis, and contributing to the stability and maturation of blood vessels [376]. These functions, first of all, are related to specific morphological organization and, secondly, to metabolic interactions between these types of cells. Morphologically, pericytes are embedded in the basal lamina of the ECs (including BECs) within capillaries and postcapillary venules [377–379]. This anatomical proximity facilitates direct cell-cell communication, mainly through the Gap junctions, positioning pericytes as principal homeostasis regulators of the local microenvironment and vessel stability [377]. Metabolic interaction between ECs (including BECs) and pericytes, based on pericytes preferential use of oxidative phosphorylation (OXPHOS) for ATP production, makes pericytes the primary stabilizers of vessel wall integrity in the vascular system [380]. Through these diverse functions, pericytes ensure the proper functioning of the vascular system and respond to various physiological and pathological conditions over the vascular bed. 

BEC morphology itself is specific and characterized by numerous caveolae, endothelial TJs, AJs, and an increased number of mitochondria [381]. The BEC surface displays dozens of specific proteins, regulating cell-cell communication and molecular transport: claudin-5, occludin 3,12 (OCCLN 3,12), Endothelial cell-selective adhesion molecule (ESAM), junction molecules 1, 3 (JAM1,3), tricellulin and lipolysis-stimulated lipoprotein receptor (LSR), vascular endothelium cadherin (VE-cadherin), N-cadherin, and transporters (Glut-1, Slc2a1) [382,383]. Notably, the BECs do not express thrombomodulin [384]. The morphology, proteome composition, and expression of CD31 and von Willebrand factor differ in various brain parts and types of vessels (capillaries, arterioles, venules) [385,386]. The local cellular environment mediates BEC functionality: microglia change BBB (blood-brain barrier) permeability also causing systemic inflammation [387], and BEC-macrophage communication instigates barrier dysfunction in patients with hypertension [388]. Plasmodium parasites initiate BBB disruption, resulting in edema [389]. Gamma interferon contributes to the BBB leakage process [390].

Endocardial ECs (EECs) display characteristics typical of continuous vascular endothelium. The EECs form a thin cell layer with varying thicknesses ranging from 50 to 300 mm. Primarily, they contribute to normal blood flow and physiological cardiac function. The total amount of EECs in the heart corresponds to 2%-3% of the heart mass [391]. These cells are characterized by the unique presence of numerous microvilli on the surface, and their basal lamina is composed of  delicate collagen and elastic fibers [392]. Compared to vascular ECs of arteries, veins, and capillaries, the EECs have a broader shape; particular cellular connections (e.g., GJs) and intercellular spaces define the specificity of these EECs. The GJs control the transendothelial permeability of EECs through a quick passage of charged ions (mostly Ca+2), messenger molecules, and small metabolites [393]. Metabolically active EECs use fatty acid and lactate as an energy source [394], and control long-chain fatty acid (LCFA) delivery and metabolite transport to myocytes [395]. GLUT-1,3,4 mediate the primary glucose uptake in EECs, while GTPases participate in cellular protein transport [1,396]. The condition of EECs is regulated by systemic and local NO synthases, endothelin -1 (ET-1), angiotensin II, prostacyclin, natriuretic peptides A and B, VEGF, hepatocyte growth factor (HGF), FGF, interleukin-6 (IL-6), thrombospondins, insulin-like growth factor -1 (IGF-1) [397,398]. EECs express various markers that are typical for all endothelial cells, including CD31, CD 34, vWf, caveolin, neuregulin-1 (NRG-1), and VE-cadherin [399]. Heart endothelium can be considered as an active paracrine, endocrine, and autocrine organ that produces cardioprotective substances like secretory leukocyte protease inhibitor (rhSLP1) [400]. Simultaneously, EECs firmly contact the cardiomyocyte surface (between the two types of cells, there is a thick fibrillary basal lamina). This cell-cell crosstalk is essential for heart regeneration and remodeling [397,401]. Release and diffusion of signaling molecules within this space participates in heart inotropy [402]. However, any extrinsic cues (factors located out of the cardiovascular system) provoke disbalance in endothelial physiology, e.g., hyperthyroidism is regarded as a factor of cardio-cerebrovascular dysfunction [403]; as well as autoimmune diseases, systemic and local inflammation can damage the morphology or physiology of vascular  ECs and EECs [404]. Hypoxia may cause upregulation of vWf in EECs through high mobility group box-1 (HMGB1) and activation of toll-like receptor-2 (TLR2), resulting in augmentation of plasma sodium concentration, increase of  E selectin and P selectin, resulting in downregulation of anti-thrombotic factors  [405] and altered innate immunity reactions [406,407].

The pulmonary ECs (PECs) contribute to the gas exchange with the external microenvironment,  formation of air-blood barrier in the alveoli, maintenance of pulmonary and systemic vascular homeostasis, immune response (through mitochondrial activation of innate immune mechanisms that stimulate lymphatic delivery to the lymphatic nodes and promotion of adaptive immunity) and provide a failsafe mechanism to balance blood pressure in the lung and possibly to regulate coagulation with PGI2 prostaglandin [408,409]. PECs are classified as vascular (macrovascular and microvascular) or alveolar ECs. Recently, Car4+high PECs (with high levels of Car4 and CD34 expression) were identified by scRNA-seq analysis. This cell subpopulation is located throughout the lung periphery and primed to respond to VEGFA signaling. The number of Car4+high PECs increases in the regenerating areas of pulmonary tissues after influenza infection [410]. The alveolar endothelium divides into two intermingled cell types: aerocytes (specialized in the gas exchange and leukocyte trafficking) and general capillary cells (functioning as stem / progenitor cells and responsible for regulating the vasomotor tone of capillaries) [411]. The pulmonary endothelial barrier plays a vital role in vascular homeostasis maintenance. TJs of  PECs are formed by occludins, claudins, and junctional adhesion molecules. Notably, primarily vascular endothelial cadherin comprises the AJs [412]. PECS is a continuous type of endothelial cells with an epithelioid shape. When metabolically active, these cells produce prostacyclin, bradykinin, angiotensin, endothelin-1, prothrombotic and anti-thrombotic factors, and other anti-inflammatory cytokines  [413,414]. PECS express vWf,  endothelial NO synthase, cadherins, CD31, and angiotensin I converting enzyme(ACEI) [415–417].

The endothelial population of the kidney ECs (KECs) is diverse. KECs include glomerular ECs (GECs) and peritubular capillary ECs (PCECs). Both types of cells have a continuous basal lamina; however, PCECs belong to the pseudo-fenestrated and GECs to the true-fenestrated ECs (Table 1) [418]. Pseudo-fenestrations of PCECs represent an incomplete fenestration with an overlying fenestral diaphragm, controlling molecular passage and exerting a “sieving” function or regulating the counter-current-based gradient in the medulla and cortex [419]. Fenestral diaphragms are typical for PCECs and intestinal ECs (but not for GECs and form open holes); they are composed of radial heparan sulfate proteoglycan fibrils (glycocalyx tufts) acting as a permselective barrier and regulating the passage of the water, small molecules [420]. Electron microscopy of GECs revealed the thick (around 200nm) glycocalyx layer [418,421,422]. GECs express vWf, vascular cell adhesion protein 1 (VCAM1), and intercellular adhesion molecule-1 (ICAM-1). GECs, together with mesenchymal cells, produce HGF, Kruppel-like factor (KLF), and insulin-like growth factor binding proteins (IGFBPs) through the activation of the c-Met receptor[423]. PCECs sit on the continuous basal lamina which separates them from the pericytes. PCECs display some immunohistochemical characteristics of macrophages: OKM5 (medullary expression only, there is no expression of OKM5 in GECs), and interleukin-1 (IL-1) expression [424,425]. In adults, PCECs display CD31 and VE-cadherin, a low level of vWf, and overexpression of the plasmalemma vesicle-associated protein 1 (PV1). GECs do not have the PV1 [426]. According to recent data, CD34  has been revealed in the peritubular microvasculature of adult human kidney, and the level of expression usually correlates with severity of  glomerular and tubulointerstitial damage [426–428].

Intestinal ECs (IECs) belong histologically to the pseudo-fenestrated endothelium with TJs and fenestral diaphragms. These cells are essential for local and general immune responses and the development of intestinal inflammation [429].  The IECs possess fenestrated diaphragms.  This fenestrated diaphragm along with the gastrointestinal mucosa, form a selective barrier  between the extracellular and intravascular space (so-called  gut-blood barrier) [430]. The healthy intestinal barrier is impermeable to 70 kDa molecules [431]. TJs and claudin proteins regulate the functionality of the intestinal microvascular endothelial barrier. The overexpression of claudin-1 increases the antiviral and antibacterial resistance of IECs by intensifying the mucosal and endothelial integrity  [432]. Downregulation of claudin-5 and claudin-8 increases barrier permeability [433]. The tight junctions belong to a dynamic structure that can adapt its protein composition according to external (pathological or physiological) stimuli [434]. The primary markers of IECs are CD31, vWf,  VE-cadherin [435] and  E-selectin. P-selectin, VCAM-1, and ICAM-1 cannot be detected in the basal, unstimulated state. However, the pro-inflammatory cytokines Il-1b and  tumor necrosis factor-a (TNF-a) can induce the biosynthesis of the latter proteins [436,437].

The liver sinusoid ECs (LSECs) are metabolically active, organ-specific ECs with great transdifferentiation potential. Phenotypically LSECs form a discontinuous fenestrated endothelium with a lacking basal lamina [438]. LSECs are organized in sieve plates and display plenty of fenestrations, ~2-20 fenestrations per μm2, corresponding to 2-10 % of the LSECs surface. The diameter and distribution of fenestrations allow rapid exchange between the space of Disse and the blood [439]. LECs are essential for: 1) the primary selective barrier (protecting liver parenchyma); 2) the formation of scavenger and endocytosis systems of the liver; 3) immune response;  4) paracrine signaling; 5) liver regeneration [440,441]. Discontinuous fenestrated LECs express such specific markers as stabilin-1, stabilin-2, liver-endothelial differentiation-association protein (LEDA-1), and CD32b [442]. Additionally, mannose receptors (CD206) and toll-like receptors are present on the surface of the LSEC, while the CD31 and vWf are not expressed in the LSCs under normal conditions and in young individuals but may be present in ECs of larger blood vessels and lymphatic vessels in the liver [443,444]. VEGFR3 is often used as a specific marker for LECs (FLT-4) [445].

Splenic sinusoidal ECs (SSECs) are the most abundant non-immune cells of the spleen which participate in forming the splenic sinusoidal wall and create a discontinuous endothelial layer adjacent to the fenestrated basal membrane. The spleen is a secondary lymphatic organ involved in immunological supervision, clearance of the blood, and maturation of immune cells [446]. The endothelium forms filamentous structures contributing to radial construction and free, yet limited and retarded blood passage [447,448]. Open blood flow is found in splenic cords [449]. The fibers around the SSECs are associated with cellular VE-cadherin, b-catenin, p120 catenin, and actin filaments [450]. Thus, SSECs, together with spleen littoral cells, specialize in the filtration of the senescent red blood cells [451]. SSECs are characterized by expression of CD31, CD8 a/a, CD271, stabilin-1, and CD206 [451,452]. The vWf is another marker of SSECs, which is upregulated by hypothermic conditions [453].

Comment 8. In the case of endothelial turnover, there is no information about quality control mechanisms, degradation processes etc.

Response: We agree with your valuable proposition and thank you for it. The necessary information has been added to the end of Section 7.2, "Endothelial Turnover, Regeneration, and Repair." Please find the supplementary information added below in red ( page 32-33 in the new version of the manuscript, lines 934-957):

 Endothelial turnover, regeneration, and repair are crucial processes for maintaining vascular health throughout life and in the post-injury period. These processes are tightly regulated to ensure proper endothelial function. Quality control mechanisms maintain the functionality of the endothelial layer, degradation processes remove damaged cells, and regeneration and repair mechanisms restore vascular integrity.

These processes must be precisely balanced; dysregulation of any of them leads to the initiation of various pathological conditions, including inflammations, cancer, cardiovascular, nervous dysfunctions, etc.

Key endothelial control mechanisms:

1)Autophagy: removing damaged organelles and proteins [511]; 2) Unfold protein response (UPR) is realized when the misfolded proteins are accumulated in the endoplasmic reticulum [512,513]; 3) DNA damage responses (DDR) are crucial for the detection and repair of DNA damage[514,515].

The degradation processes consist of 1) ubiquitin-proteasomes system (UPS) (tags damaged or unnecessary proteins with ubiquitin for degradation) [516,517]; 2) lysosomal degradation (breaks down cellular waste) [518]; 3) caspase-mediated degradation (activated during apoptosis)[519].

Regeneration mechanisms are composed of:  1) proliferation of existing ECs[520,521]; 2) transdifferentiation (cells change from one type to another and replenish the endothelial layer)[522]; 3) recruiting of circulating cells(circulating cells differentiate into mature ECs to aid in repair)[523].

Repair mechanisms include 1) wound healing (natural response to injury) [524]; 2) shear stress active adaptation (ECs adapt to changes in blood flow)[525]; 3) EndMT ( ECs transform into mesenchymal cell to aid in tissue repair and remodeling)[526].

Reviewer 2 Report

Comments and Suggestions for Authors

This manuscript contains an impressive amount of cell biological information on endothelial cells. I consider it as a standard state-of-the-art source for scientists who are studying the vasculature both of blood vessels and lymph vessels. 

The reference list contains almost 500 references of which approx. 25% (over one hundred articles) have been published in the period 2020-2024, so it is pretty much an update.

I have only a few small comments that may be considered by the authors to include in a revised version of the manuscript, as follows:

1. I find the title not attractive enough to sufficiently draw the attention of scientists active in tne field. Something like 'Dynamics of endothelial cell diversity and plasticity in health and disease' would be more attractive.

2. The amount of abbreviations is spectacular and thus a list of abbreviations is necessary to facilitate reading the text. For example, it took me quite a while to discover what AG stands for (angiogenesis), and this happened quite frequently.

3. The role of pericytes in the vessel walls and in particular in barrier endothelium is not explained properly. As far as I could discover, they are only mentioned in lines 444-448. There is a lively exchange of molecules such as metabolites like lactate exchanged between ECs and PCs (see eg PMID 38733294). It means that interactions between these 2 cell types in vessel walls are an important aspect of the functioning of ECs.

4. In line 56, the outdated and incorrect term 'basement membrane' is used together with the correct term 'basal lamina', whereas in the manuscript and figures 'basal lamina'is used. Unfortunately 'basement membrane' is still used very frequently although the authors are fully correct that it is extracellular matrix. So, either leave out the term or explain that it is a wrong term.

5. In lines 44-47, high endothelial venules are not considered, which are essential for immune cells to leave the circulation to enter surrounding tissues. They should be mentioned besides all other phenotypes of ECs.

6. In the reference list a substantial amount of references such as 9, 10 and 14 include a phrase like [cited 2024 May 17]. What is meant by that as the respective references have been published in earlier years. Please, explain.

 In conclusion, my comments are small issues and should be considered as a positive gesture to the authors who have written an impressive review. I am pretty convinced that it will be cites frequently.

Author Response

Dear Colleague, Dear Reviewer 2,

We would like to thank you for your attention to our review. We found your comments to be essential and extremely helpful in improving our work. Your insights have provided us with valuable guidance, and we appreciate the time and effort you have invested in reviewing our manuscript.

Comment 1. I find the title not attractive enough to sufficiently draw the attention of scientists active in the field. Something like 'Dynamics of endothelial cell diversity and plasticity in health and disease' would be more attractive.

Response: We agree with and accept your valuable proposition. The title of the article has been changed in the new version of the manuscript to “Dynamics of Endothelial Cell Diversity and Plasticity in Health and Disease” (page 1, line 2).

Comment 2. The amount of abbreviations is spectacular and thus a list of abbreviations is necessary to facilitate reading the text. For example, it took me quite a while to discover what AG stands for (angiogenesis), and this happened quite frequently.

Response: Thank you for this suggestion. We agree with your comment and have made the following changes:

   1) Many abbreviations have been removed from the text.

   2) The retained abbreviations are now accompanied by their full terms at their first mention, e.g., intraembryonic hemogenic endothelial cells (IHECs), fibroblast growth factors (FGFs), etc.

    3) We have added a list of abbreviations at the end of our review (page 33 in the manuscript).

ABBREVIATIONS

ANG1 and ANG 2-angiopoietins 1 and 2

ACE1-Angiotensin 1-converting enzyme

BMP- bone morphogenetic proteins pathway

DNA-deoxyribonucleic acid

ECM-extracellular matrix

ECs-Endothelial cells

EMT - epithelial-mesenchymal transition

EndMT-endothelial-to-mesenchymal transition

ER71 (ETV2) - ETS variant transcription factor 2

ESAM-Endothelial cell-selective adhesion molecule

ETS family –E twenty-six family transcription factors

FGF -fibroblast growth factor

FLI1-friend leukemia integration-1 transcription factor

HES1, 2-hairy and enhancer of split 1, 2

HEY1-hairy/enhancer of split related with YRPW motif protein 1

HGF- hepatocyte growth factor

HIF-1a- hypoxia-inducible factor 1a

Hh -Hedgehog

ICAM-1 -intercellular adhesion molecule-1 (or CD54)

IHECs-intraembryonic hemogenic endothelial cells

LMO2-Lim domain only 2

LSECs-liver sinusoid ECs

LYVE 1-lymphatic vessel endothelial hyaluronan receptor

MAML1,2,3- mastermind like protein 1,2,3

MAP4K4 -mitogen-activated protein- 4- kinase 4

NRP 1,2 -neuropilin1, 2

PI3K/Akt phosphoinositide 3-kinase / protein kinase B

PDGF - Platelet-derived Growth factor

PDGFR-b-Platelet-derived Growth factor receptor-b

RNA-ribonucleic acid

RUNX1-runt-related transcription factor 1

S1PR1sphingosine 1-phosphate (S1P)- sphingosine 1-phosphate receptor

SCL/TAL1 or TAL1 or SCL- stem cell leukemia/or T-cell acute lymphocytic leukemia-1

scRNA-seq analysis – single-cell RNA sequencing analysis

SFRP-secret frizzled-related protein

SMADs homologies to SMA ("small" worm phenotype) and MAD family ("Mothers Against Decapentaplegic") genes

SSECs-spleen sinusoid ECs

TAK-1 -transforming growth factor-beta activated kinase 1

TCF7L2-transcription factor 7 like 2 (TCF4)

TGFb ,a -transforming growth factor b

TIE-tyrosine kinase with immunoglobulin-like and EGF-like domains

TLR2 -toll-like receptor-2

TNF-a-tumor necrosis factor-a

TWIST1 -twist family of basic helix-loop-helix protein 38 (bHLHa38) transcription factor 1

VCAM1-vascular cell adhesion protein1

VE-cadherin -vascular endothelial cadherin (cadherin-5 or CD144)

VEGF -vascular endothelial growth factor

VEGFR-1 (Flt 1)-vascular endothelial growth factor receptor 1

VEGFR2 (Flk1/KDR) -vascular endothelial growth factor receptor 2

vWf -von Willebrand factor

WNT- Wingless-related integration site

Comment 3. The role of pericytes in the vessel walls and in particular in barrier endothelium is not explained properly. As far as I could discover, they are only mentioned in lines 444-448. There is a lively exchange of molecules such as metabolites like lactate exchanged between ECs and PCs (see eg PMID 38733294). It means that interactions between these 2 cell types in vessel walls are an important aspect of the functioning of ECs.

Response: Agree. An additional paragraph about the role of pericytes in the vessel walls and their relationship with endothelial cells (ECs) was added in paragraph 6, on page 28, lines 689-705, in the new version of the manuscript. The additional text is highlighted in red.

  1. Selected tissue-specific endothelial phenotypes

The brain, heart, lungs, liver, kidneys, gut, and endocrine organs have distinct endothelial subpopulations displaying specific characteristics.

Brain endothelial cells (BECs), along with pericytes and astrocytes, form the blood-brain barrier and enforce the protective barrier function [374]. The relationships between pericytes and ECs (including BECs) are dynamic and vital for the integrity and maintenance of vessel walls [375]. Functionally, pericytes play a vital role in modulating blood flow, supporting angiogenesis, and contributing to the stability and maturation of blood vessels [376]. These functions, first of all, are related to specific morphological organization and, secondly, to metabolic interactions between these types of cells. Morphologically, pericytes are embedded in the basal lamina of the ECs (including BECs) within capillaries and postcapillary venules [377–379]. This anatomical proximity facilitates direct cell-cell communication, mainly through the Gap junctions, positioning pericytes as principal homeostasis regulators of the local microenvironment and vessel stability [377]. Metabolic interaction between ECs (including BECs) and pericytes, based on pericytes preferential use of oxidative phosphorylation (OXPHOS) for ATP production, makes pericytes the primary stabilizers of vessel wall integrity in the vascular system [380]. Through these diverse functions, pericytes ensure the proper functioning of the vascular system and respond to various physiological and pathological conditions over the vascular bed. 

BEC morphology itself is specific and characterized by numerous caveolae, endothelial TJs, AJs, and an increased number of mitochondria [381]. The BEC surface displays dozens of specific proteins, regulating cell-cell communication and molecular transport: claudin-5, occludin 3,12 (OCCLN 3,12), Endothelial cell-selective adhesion molecule (ESAM), junction molecules 1, 3 (JAM1,3), tricellulin and lipolysis-stimulated lipoprotein receptor (LSR), vascular endothelium cadherin (VE-cadherin), N-cadherin, and transporters (Glut-1, Slc2a1) [382,383]. Notably, the BECs do not express thrombomodulin [384]. The morphology, proteome composition, and expression of CD31 and von Willebrand factor differ in various brain parts and types of vessels (capillaries, arterioles, venules) [385,386]. The local cellular environment mediates BEC functionality: microglia change BBB (blood-brain barrier) permeability also causing systemic inflammation [387], and BEC-macrophage communication instigates barrier dysfunction in patients with hypertension [388]. Plasmodium parasites initiate BBB disruption, resulting in edema [389]. Gamma interferon contributes to the BBB leakage process [390].

Commnet 4. In line 56, the outdated and incorrect term 'basement membrane' is used together with the correct term 'basal lamina', whereas in the manuscript and figures 'basal lamina'is used. Unfortunately 'basement membrane' is still used very frequently although the authors are fully correct that it is extracellular matrix. So, either leave out the term or explain that it is a wrong term.

Response: Agree. The term "basement membrane" was replaced with "basal lamina" throughout the text.

Page 9- table2 ; Page 24- lines 519, 523, 531, 535;Page25-line 547, Page 27-line 649; line 696 page 28, Page 29 , line 724 etc

Comment 5. In lines 44-47, high endothelial venules are not considered, which are essential for immune cells to leave the circulation to enter surrounding tissues. They should be mentioned besides all other phenotypes of ECs.

Response: Agree. A paragraph about high endothelial venules and endothelial cells was added in paragraph 5.1, "Diversity of ECs in the Microcirculation," on page 26, lines 566-577, in the new manuscript.

Glomerular fenestrated ECs take the intermediate position between common pseudo-fenestrated and discontinuous fenestrated ECs: they have a basal lamina, pores with a diameter 60-80, and no diaphragms (Table 2) [303,311,312]. The local microcirculatory functions and morphology contribute to permeability regulation, i.e. an exchange between the vascular lumen and the extravascular tissue [269,313,314]. The above-mentioned phenotypic differences of ECs directly affect microcirculatory endothelial permeability [315].

High endothelial venules (HEVs) are a special part of the microcirculatory bed[316]. These vessels belong to post-capillary venules, which mostly accompany lymphoid tissues such as lymph nodes, Peyer’s patches, and tonsils. The specificity of these cells is strongly related to the immune system functioning and serves to facilitate the entry of lymphocytes from the bloodstream into the lymphoid tissues( place of immune response initiation and regulation). The ECs of  HEVs have a flat and cuboidal shape, unlike the flat and elongated morphology of typical ECs. Notably, the surface of these types of ECs displays a specific glycosylation pattern composed of  Pereferial node addressins (PNAd) and express high levels of sialomucins which contribute to lymphocyte homing through binding L-selectin on lymphocytes[317]. The listed structural and functional characteristics facilitate the mediation of lymphocyte trafficking into lymphoid tissues, supporting effective immune surveillance and responses[318].

Critical structural elements of the endothelia, regulating the cellular diffusion process (especially across the continuous endothelium) and layer stability, are cell-to-cell junctions [319]. Junctional endothelial diversity is another marker of EC heterogeneity, characterized by variations in the expression of junctional proteins[320]. The junctions connect to cytoskeletal and several signaling proteins that contribute to the maintenance of shape and polarity [321,322]. There are three types of cellular junctions: tight junctions (TJs), adherence junctions (AJs), and gap junctions (GJs) [278].

Comment 6. In the reference list a substantial amount of references such as 9, 10 and 14 include a phrase like [cited 2024 May 17]. What is meant by that as the respective references have been published in earlier years. Please, explain.

Response: Agree. Thank you for your detailed approach to reference control. The explanation is very simple: we forgot to switch off the option for the last citation in our reference management engine. Our review was revised several times, with adaptations to the reference list. [Cited 2024 May 17]. This means that this citation ( last version)  was added or revised in the manuscript on May 17, 2024. The necessary corrections have been made. We apologize for the inconvenience.
